



# The benefits of increasing resolution in global and regional climate simulations for European climate extremes

Carley E. Iles[1], Robert Vautard[1], Jane Strachan[2], Sylvie Joussaume[1], Bernd R. Eggen[2] and Chris D. Hewitt[2]

[1] Laboratoire des Sciences du Climat et de l'Environnement, LSCE-IPSL, CEA-CNRS-UVSQ, Université Paris-Saclay, F-91198 Gif-sur-Yvette, France
[2] Hadley Centre, Met Office, Exeter, UK

*Correspondence to*: Carley E. Iles (carley.iles@lsce.ipsl.fr)

**Abstract.** Many climate extremes, including heatwaves and heavy precipitation events, are projected to worsen under climate change, with important impacts for society. Future projections, required for adaptation, are often based on climate model simulations. Given finite resources, trade-offs must be made concerning model resolution, ensemble size and level of model complexity. Here we focus on the resolution component. A given resolution can be achieved over a region using either global climate models (GCMs) or at lower cost using regional climate models (RCMs) that dynamically downscale coarser GCMs. Both approaches to increasing resolution may better capture small-scale processes and features (downscaling effect), but increased GCM resolution may also improve the representation of large-scale atmospheric circulation (upscaling effect). The size of this upscaling effect is therefore important for deciding modelling strategies. Here we evaluate the benefits of increased model resolution for both global and regional climate models for simulating temperature, precipitation and wind extremes over Europe at resolutions that could currently be realistically used for coordinated sets of climate projections at the pan-European scale. First we examine the benefits of regional downscaling by comparing EURO-CORDEX simulations at 12.5 and 50 km resolution to their coarser CMIP5 driving simulations. Secondly, we compare global scale HadGEM3-A simulations at three resolutions (130, 60 and 25 km). Finally, we separate out resolution dependent differences for HadGEM3-A into downscaling and upscaling components using a circulation analogue technique. Results suggest limited benefits of increased resolution for heatwaves, except in reducing hot biases over mountainous regions. Precipitation extremes are sensitive to resolution, particularly over complex orography, with larger totals and heavier tails of the distribution at higher resolution, particularly in the CORDEX vs CMIP5 analysis. CMIP5 models underestimate precipitation extremes, whilst CORDEX simulations overestimate compared to E-OBS, particularly at 12.5 km, but results are sensitive to the observational dataset used, with the MESAN reanalysis giving higher totals and heavier tails than E-OBS. Wind extremes are somewhat stronger and heavier tailed at higher resolution, except at coastal regions where large grid boxes spread strong ocean winds further over land. The circulation analogue analysis suggests that differences with resolution for the HadGEM3-A GCM are primarily due to downscaling effects.



## 1 Introduction

Climate extremes, such as heatwaves and heavy precipitation events are projected to worsen under climate change, with important impacts for society (Seneviratne et al., 2012). Such projections are generally based on numerical climate model simulations. However, given finite computational resources, trade-offs between model resolution, ensemble size and the level of model complexity are necessary. For extreme events driven by large-scale processes such as long-standing anticyclones, the proper simulation of the amplitude of extremes is limited by dynamics but also by land-atmosphere feedbacks and the many physical processes involved in the surface energy budget. Such extremes are typically heat waves, droughts and cold spells. Many other types of extreme event are by nature small scale. Such is the case of convective precipitation, floods, extratropical wind storms, cyclones and medicanes. The resolution of Global Climate Models (GCMs) in CMIP5 (Coupled Model Intercomparison Project Phase 5; Taylor et al., 2012) does not allow these events to be resolved explicitly. Increased resolution in GCMs may improve the representation of small-scale processes and features, including orography and coastlines (downscaling effect), but potentially may also improve the representation of the interaction between small and large scale dynamical processes and ultimately improve the large-scale atmospheric flow (upscaling effect). For instance, a better representation of baroclinic eddies may help to better simulate large Rossby waves such as those inducing long-lived anomalies, due to the inverse energy cascade. This may improve the simulation of the frequency and duration of heat waves and cold spells, and related anomalies such as summer droughts. For precipitation and wind extremes, an improvement with resolution could be expected due to the small-scale processes and features involved. However, upscaling effects may also have benefits by improving storm-track location, and duration of wet spells. An alternative approach to increasing the resolution of global-scale models is to use regional climate models (RCMs) driven by coarser GCMs to achieve a given high resolution over a limited area at lower cost. However, this technique only captures downscaling effects, since the RCM inherits the large scale circulation from the driving GCM.

Current generation GCMs commonly used for climate projections (e.g. CMIP5 models) have a resolution ranging from about 70 to 250 km resolution, although 25 km GCMs are starting to be run under projects such as PRIMAVERA and HighResMIP (part of CMIP6; Haarsma et al., 2016). For coordinated RCM experiments, such as CORDEX (Coordinated Regional Downscaling Experiment; Giorgi et al., 2009), resolutions are generally between 10 to 50 km (e.g. Jacob et al., 2014). In order to simulate convective precipitation a resolution of <5 km is really needed, which is very computationally expensive, but such ensembles of convection permitting RCMs are currently in development (Coppola et al., 2019). An important question is the extent to which increased resolution benefits the simulation of extreme events for both global and regional models for the kind of resolutions that can realistically be run for coordinated pan-continental climate projections. Particularly, whether using global high resolution adds further benefits over regional high resolution due to an improved large scale circulation. We will address these questions focusing on Europe, whose climate is highly variable and affected by a range of both large and small scale processes, which present challenges for adequate simulation. We focus on extreme precipitation, temperature and wind, to cover a range of events that may be affected by resolution in different ways.





The benefits of increased resolution for European precipitation extremes are well documented, whilst the effects on heatwaves, cold spells and wind extremes are less well known. In GCMs, global precipitation tends to increase with resolution, and for grid point models the fraction of land precipitation and moisture fluxes from land to ocean increases, largely due to better resolved orography (Vannière et al., 2019; Terai et al., 2018; Demory et al., 2014). Precipitation extremes tend to get heavier and agree better with observations (Wehner et al., 2010, O'Brien et al., 2016; Kopparla et al., 2013; Shields et al., 2016; Vannière et al., 2019), unless the parameterisation schemes are not suited to the resolution (e.g. Wehner et al., 2014). In Europe, Schiemann et al. (2018) find that both mean and extreme precipitation are simulated better with increased resolution in HadGEM3A, mostly originating from better resolved orography. In contrast, Van Haren et al. (2015a) find that improvements in Northern and Central European mean and extreme winter precipitation with resolution are mostly associated with improved storm tracks in EC-Earth. For RCMs, extreme precipitation is improved with resolution when compared to high resolution observations, particularly over orography, including frequency-intensity distributions and spatial patterns, (see e.g. Torma et al., 2015 and Prein et al., 2016 for EUROCORDEX at 12.5 km vs 50km and vs the driving GCMs, and Ruti et al., (2016) for Med-CORDEX). However, benefits are smaller for regional and seasonal mean precipitation. Convection permitting models (<4km resolution) are particularly beneficial in simulating summer extreme and sub-daily precipitation, including the diurnal cycle of convection, but can overdo extreme precipitation (e.g. Prien et al., 2015; Kendon et al., 2012; 2014).

For heatwaves, increasing resolution does not lead to obvious benefits in RCM simulations (see e.g. Vautard et al., 2013 for EURO-CORDEX), except improved spatial detail (Gutjahr et al., 2016). However, increased resolution may have more impact in global models since the large scale circulation that contributes to their formation may be affected. This remains a largely unstudied question, with the exception of a few studies such as Cattiaux et al. (2013) who find that increasing resolution in the IPSL GCM leads to a reduction in the cold bias of both cold and warm extremes in Europe, along with improved statistics, such as duration and frequencies and improved weather regimes.

For wind extremes, stronger winds and better spatial detail with resolution have been found for regional models (e.g. Pryor et al., 2012; Kunz et al., 2010). Donat et al. (2010) found that observed storm loss estimates for Germany could be reconstructed more accurately through dynamical downscaling compared to using the coarser resolution driving ERA-40 data directly. Ruti et al., (2016) found improvements in Mediterranean cyclogenesis in coupled Med-CORDEX RCMs relative to the ERA-interim driving data, whilst extreme winds over the Mediterranean generally improve (i.e. are stronger) with higher resolution RCMs (e.g. Ruti et al. 2016; Hermann et al. 2011). However, most GCM studies focus on the simulation of extratropical cyclones rather than wind directly. Such studies find an improvement in the representation of various aspects of Northern Hemisphere extratropical cyclones with increased resolution, including frequency, intensity and the position of the storm tracks (Colle et al., 2013; Jung et al., 2006; 2012), even in the higher resolution CMIP5 models (~<130 km; Zappa et al., 2013). Vries et al., (2019) found that the resolution of Atlantic Gulf-Stream SST fronts affects winter extratropical cyclone strength. Whether these improvements translate into an improvement in wind extremes remains to be assessed.



Persistence of weather regimes, such as blocking or the phase of the North Atlantic Oscillation, can be important
drivers for extreme events in Europe. Using the ECMWF IFS model, Dawson et al., (2012; 2015) find that such weather
regimes cannot be simulated realistically at typical CMIP5 resolution (~125 km), but are improved at 40 km, and well-
simulated at 16km. Cattiaux et al., (2013) find improvements at more modest resolutions in the IPSL model. Blocking
frequency tends to be underestimated by CMIP5-resolution climate models (Anstey et al., 2013). This tends to be
improved with resolution, particularly over the North Atlantic (Jung et al., 2012, Anstey et al., 2013; Matsueda et al.,
2009, Berckmans et al., 2013, Davini et al., 2017a; 2017b), although results tend to be somewhat sensitive to season
and model considered (Schiemann et al., 2017) and compensating errors may be involved (Davini et al., 2017a for EC-
EARTH). O'Reilly et al. (2016) find that having a well-resolved Gulf stream SST front is also important for European
winter blocking and associated cold spells. An important question is whether these improvements in the large scale
circulation translate into an improvement in the simulation of European climate extremes.
Here we examine the benefits of increased resolution for global models compared to regional models for the simulation
of European heatwaves, heavy precipitation events and wind storms. We further break down any resolution related
differences for a global model into upscaling and downscaling components. This will shed light on whether potential
improvements in the large scale circulation suggested in the literature translate into an improved representation of
climate extremes. This is an important consideration in choosing how to distribute finite resources between global and
regional models. We focus on the kind of models widely used to provide climate projections at a European scale,
applying a consistent approach across model types. Firstly, the benefits of regional dynamical downscaling are
explored by comparing EURO-CORDEX simulations at 50 and 12.5 km resolutions to their coarser driving CMIP5
GCMs. Secondly, the benefits of increased resolution for a global model are examined using HadGEM3-A at 130, 60
and 25 km resolution. Finally, the roles of upscaling versus downscaling will be examined using a circulation analogue
technique applied to HadGEM3-A.
**2 Observational and model data**
**2.1 Observations**
Model simulations are evaluated using observational datasets. For daily precipitation and daily maximum temperature,
we use the gridded station based dataset E-OBS on a 0.5° latitude-longitude grid (Haylock et al. 2008). This covers
the European domain from 1950 to present. Gridded datasets tend to reduce the magnitude of extremes compared to
station data through smoothing effects, but are more comparable to the grid box averages from GCMs (Haylock et al.
2008). Nevertheless, E-OBS has a relatively low underlying station density, and tends to underestimate precipitation
extremes relative to higher density regional datasets), due to missed extremes which are local in scale (Prein and Gobiet
2017). However, such high resolution datasets are not available at a pan-European scale. As a compromise, results are
repeated for precipitation extremes using the MESAN reanalysis (Landelius et al. 2016), which combines information
from the high resolution HIRLAM reanalysis (Dahlgren et al. 2016) with a network of station-based observations. For
much of Europe these are the same as those used for E-OBS, but with the addition of Swedish Meteorological and





Hydrological Institute (SMHI) stations over Sweden, and a high density of Meteo-France stations over France
(Landelius et al. 2016). MESAN provides daily precipitation data for the more limited period 1989-2010. We use the
version available on a 0.11° rotated grid. Prein and Gobiet (2017) find that it gives heavier extremes than E-OBS in
some regions (France, Spain, the Carpathians), but generally not as high as the high resolution regional datasets (except
in France). Neither dataset is corrected for gauge undercatch, which tends to be around 3-20% for rain, and up to 40%
for snow, or even 80% for non-shielded gauges (Førland and Institutt 1996; Goodison et al. 1997).

Wind extremes tend to happen on sub-daily time scales, necessitating the use of sub-daily data to avoid missing as
many events (although events, or their peak magnitude, will still be missed). We use three observational wind datasets.
These are all based on the ERA-Interim reanalysis (Dee et al. 2011), but differ in the way they are processed. The first
is the WFDEI dataset (WATCH-Forcing-Data-ERA-Interim; Weedon et al. 2014) and the other two are ECEM datasets
(European Climate Energy Mixes; Jones et al. 2017). One ECEM dataset is bias corrected using a Weibull distribution
based on the HadISD station dataset (Dunn et al. 2012) applied to each grid cell (ECEM-wbc), whereas the other
version contains no bias correction (ECEM-noc). WFDEI is available at 3 hourly resolution, whereas ECEM is 6
hourly. Therefore, 6 hourly data is used from both datasets for consistency. All datasets are available on a 0.5° regular
latitude longitude grid for the period 1979-2016. Although neither ECEM-noc or WFDEI are bias corrected, they
nevertheless give different values, presumably due to differences in interpolation method from the original 0.7° grid
of ERA-Interim.
**2.2 Climate model data**
**2.2.1 EURO-CORDEX and CMIP5**
In order to examine the effect of dynamical downscaling for climate extremes, we make use of the EURO-CORDEX
(Jacob et al. 2014) RCM simulations for the historical period over the European domain which are driven by lower
resolution global scale coupled CMIP5 GCMs. The GCMs are forced by observed records of anthropogenic and natural
forcings, such as greenhouse gases, anthropogenic aerosols, land use changes, solar variability and volcanic aerosols
to allow comparability to historical records. For the most part the RCMs inherit the effects of these forcing agents from
the GCMs, with the exception of greenhouse gases, which are prescribed. A comparison of the RCM simulations with
their driving CMIP5 simulations allows us identify any value added by regional high resolution. The EURO-CORDEX
simulations are available at 0.11° and 0.44° (12.5 km and 50 km respectively), allowing an assessment of the difference
that increased regional resolution brings. By examining the subset of GCM-RCM combinations that are common to
both CORDEX resolutions along with their driving GCMs we can isolate the effects of changing resolution.

Daily precipitation (pr), daily maximum temperature (tasmax), and daily maximum surface wind speed (sfcWindmax)
were taken from both CORDEX and CMIP5. The simulations used are shown in Table S1. This consists of 23 and 19
simulations for precipitation for the 0.44 and 0.11 simulations respectively, with 15 common to both categories with
data also available from their driving GCMs (from now on referred to as "common to all" or "common subset"); 22
and 18 respectively for temperature, with 14 common to all, and 15 and 14 for wind with 6 that are common to all. We





also extend the analysis to all other historical CMIP5 GCMs with the relevant variables, with 126 simulations from 41
GCMs for precipitation, 115 from 39 models for temperature and, 61 simulations from 28 models for wind. For wind,
using 3 or 6 hourly data would have made results more comparable to the observational wind datasets (see above).
However, such data were not available for the 0.44° CORDEX simulations, and for only three CORDEX simulations
at 0.11° resolution which also had data for their driving GCMs, all three of which use the same RCM (RCA). We
therefore use the variable sfcWindmax (daily maximum surface wind speed) which was available for many models.
The implications of this are discussed further in the results section.

### 2.2.2 UPSCALE simulations

In order to examine the benefits or otherwise of differences in resolution for a global model, we make use of simulations
undertaken as part of the UPSCALE project (UK on PRACE: weather-resolving Simulations of Climate for globAL
Environmental risk; Mizielinski et al. 2014). This consists of the atmosphere only version of the Hadley Centre Global
Environment Model 3 (HadGEM3-A) run at three different resolutions: N96 (130 km), N216 (60 km) and N512 (25
km), all with 85 vertical levels for the period 1985-2011, with 5, 3 and 5 ensemble members respectively (or 3, 3 and
5 for wind data). The simulations are forced by observed records of greenhouse gases, aerosols, ozone, solar variability
and volcanic forcings following the AMIP-II procedure (Taylor et al. 2000), and an alternative dataset for sea surface
temperatures (SSTs) and sea ice. Very few parameters differ between the resolutions, enhancing the comparability of
the three ensembles. We use daily precipitation data, daily maximum temperatures and 3-hourly wind (subsampled to
6-hourly).

### 2.2.3 Regridding

In order to compare models of different resolutions with each other and with observations it was necessary to regrid
variables to a common grid. We use a 0.5° regular longitude-latitude grid since it is the resolution of the majority of
the observational datasets used (E-OBS, ECEM and WFDEI) and is computationally feasible. Some of the benefits of
higher resolution may be lost by doing this, putting our results on the conservative side. However, sensitivity tests
showed that results for MESAN did not change perceptibly by using a 0.5° grid as compared to a 0.1° regular grid
(chosen to be close to the original 0.11° rotated grid). We regrid the daily data, before the calculation of annual extreme
indices.

Sensitivity of results to regridding technique was investigated for precipitation and wind for a number of models of
different resolutions and compared with results based on using the original grids (Figure S1). For the coarser resolution
models (e.g. HadCM3) results for precipitation extremes were very sensitive to regridding technique, with much
weaker extremes for some techniques e.g. distance-weighted average remapping and bilinear interpolation, with
unrealistic artefacts in the spatial patterns for many methods. For high resolution models, regridding technique did not
make much difference to results. Overall the nearest neighbour method was chosen since it gave results very close to
using the original grid for all model resolutions, preserving the amplitude of extremes, and also having minimal
artefacts when plotting spatial patterns of precipitation extremes. Whilst nearest neighbour may not be the best choice





in regridding from high resolution to lower (e.g. for MESAN and CORDEX 0.11), since information from only a
subset of grid cells is incorporated, results were the same when repeated using bicubic remapping. Results for wind
for the coarser models were also sensitive to regridding technique; the nearest neighbour method was again chosen
since it also performed well here, both in terms of minimising artefacts and replicating results using the original grid.
For temperature, which tends to be more uniform over large areas, bilinear interpolation was used, since the choice of
regridding technique is anticipated to be less important.

## 219    3 Methods

### 220    3.1 Extremes Indices

In order to examine extremes, we adopt indices based on the ETCCDI indices (Zhang et al. 2011). For precipitation
these are the annual maximum daily precipitation (Rx1day) and the annual maximum consecutive 5-day total
(Rx5day). For temperature we use the annual maximum daily maximum temperature (TXx) and the annual maximum
consecutive 5-day mean of daily maximum temperature (TXx5day). For wind we use the annual maximum of daily
maximum wind, which we refer to as (WindXx). This is based on sfcWindmax for the CMIP5 and CORDEX models,
and on 6-hourly data for the UPSCALE simulations and the observational wind datasets. These are therefore much
more rare extremes than those based e.g. on the 95[th] or even 99[th] percentile which would happen on average 1 in 20
days and 1 in 100 days respectively.

In order to examine how well the climate models simulate extremes and the differences between different resolutions,
we first examine the spatial patterns of the climatological mean values of the indices and their biases with respect to
observations. We then examine return period plots (see definitions below) for a number of regions for each index,
which highlights any differences in the shape of the tails of the distribution of the extremes. The regions used are based
on the PRUDENCE regions (Christenson and Christenson 2007) and the IPCC SREX regions (Seneviratne et al. 2012)
and are shown in Figure S2 and Table S2. A subset of representative regions are presented here, with some comments
about the others.

### 237    3.2 Return periods

In order to calculate regional return periods and return values we first sort the data into ascending order for each grid
cell, and then calculate the area weighted regional averages. The return periods are calculated as N/k where N is the
number of years of data, and k is the rank, with k=1 for the largest value. Return periods are therefore the inverse of
the probability of an event exceeding a given value (called the "return value"). The regional average is made, for given
return periods, over the associated return values. To avoid complications from missing data, grid cells in E-OBS with
more than 5 days of missing data in any year during the period examined were masked for the whole period. Having
one or more years missing would complicate the calculation of regional mean return periods and values. Models and
observational datasets are masked to have the same spatial coverage, which is land only. A common time period, across
the models being examined and the observations they are being compared to, are chosen to allow comparability. For


the CMIP5 and CORDEX analysis 1970-2005 is used for temperature and precipitation and 1979-2005 for wind. For
the UPSCALE runs we use 1985-2011 for temperature, and 1989-2010 for precipitation to allow comparisons with
MESAN (1986-2011 is used for the analogue analysis, see below) and 1986-2011 for wind.

Return values and periods are also calculated for the "pooled ensemble". For each grid cell, all simulations of a certain
type are combined into one long time series before being sorted into ascending order, and then regional means are
calculated as above. The models are first bias adjusted by subtracting the difference between their climatology of the
index in question and the climatology of the observations at a grid cell level. This adjustment avoids, for example,
models with particularly hot extremes dominating the ends of the tails of the distributions and allows differences in
the shapes of the distribution tails of different models to be compared more easily. Figure S3 shows the resulting spread
of models across the distributions.

In order to allow comparability of results between the EURO-CORDEX ensembles at both resolutions and their driving
CMIP5 GCMs, we picked a subset of models that are consistent across each category; that is the same GCM-RCM
combinations are used across both the 0.11 and 0.44° CORDEX categories, and are compared to the CMIP5 model
runs that were used to drive them (Table S1). We refer to these simulations as the "common subset" (see section 2.2.1).
The only exception is that the EC-EARTH ensemble member "r3" was not available for download from ESGF, so r2
was substituted instead. Since more than one EURO-CORDEX RCM is driven by the same ensemble member of the
same GCM, we repeat these GCMs when calculating the CMIP5 ensemble mean and pooled results for the common
subset. For the UPSCALE simulations, since the same version of the same model is used across each resolution, results
can also be examined without bias adjusting the extremes climatology, and this provides some interesting insights.

Confidence intervals are calculated using a bootstrapping method. If, for example, the analysis period was 1970-2005
(i.e. 36 years), 1000 random samples of 36 years from this period are chosen from the same observations/ simulation(s),
allowing the same year to be chosen more than once per iteration. For each random sample, the chosen values are
sorted for each grid cell and a regional average is calculated as above, effectively yielding 1000 return period curves
per region. The $5^{th}$ and $95^{th}$ percentile of these values are then calculated to give the confidence intervals.

Finally, for the HadGEM3-A GCM simulations, a circulation analogue technique is used to split any differences in
results according to resolution into upscaling (i.e. improved large scale circulation) and downscaling effects. This is
described in section 4.3.





## 4 Results

### 4.1 The benefits of regional high resolution: EURO-CORDEX versus CMIP5

#### 4.1.1 Temperature extremes

Figure 1 shows the spatial patterns of the climatological mean of TXx5day for the period 1970-2005 for E-OBS, and the multi-model means (MMM) of CMIP5, and CORDEX at both resolutions, along with their biases with respect to E-OBS. The first two columns are based on a subset of CORDEX simulations that use the same GCM-RCM combinations at both resolutions, whilst the CMIP5 MMM is based only on the CMIP5 simulations that drive these RCMs, with repetition of the GCMs that drive more than one RCM. The last two columns are based on the mean of all available simulations for each category to check how representative the results based on the subset are of the whole ensembles. The same general pattern can be seen in both the observations and the models, with hotter extremes in the south and cooler extremes in the north and over the mountains. At higher resolution the colder extremes over the Alps and Carpathians become more distinct. For the "common subset" the pattern of biases relative to E-OBS is similar for each model category with cold biases in the North and West and hot biases in the South-East. However, the hot biases over the mountains reduce with higher resolution since the model topography is higher. The cold bias over Scandinavia is also larger in CORDEX than in CMIP5. Biases using the whole ensemble are very similar as those for the CORDEX subset, although for CMIP5 the hot biases over the south-east, and over mountain ranges are stronger. Findings for TXx are similar, but hotter (not shown).

To give an idea of the level of consistency of results between models, results for individual models are shown in figure S4. Although the CMIP5 models agree on the general spatial pattern of temperature extremes, their absolute magnitudes vary considerably, although all are too hot over the Alps. There are also substantial differences between results from different RCMs, including those driven by the same GCM. Biases of individual RCMs do not appear systematically smaller than that of their driving GCM. Patterns are very similar for the same GCM-RCM chains at the both 12.5 and 50 km resolutions. Results for different ensemble members of the same GCM or GCM-RCM chain are very consistent, suggesting that the differences between models are not due to internal variability.

In order to assess the shape of the statistical distribution of temperature extremes, figure 2 (left column) shows return period against magnitude for TXx5day for CMIP5, CORDEX at both resolutions and E-OBS, for individual models (thin lines) and the pooled ensembles (circles) both for the common subset of models (darker circles) and all models (lighter circles) (see Methods). Results are shown for Northern, Central and Southern Europe, and are representative of the subregions. There is no obvious difference in the shape of the tails between CMIP5 and CORDEX, apart from marginally heavier tails for CORDEX 0.44 in central Europe. Agreement with E-OBS is good for the pooled ensemble, although many individual ensemble members lie outside the range of the observational uncertainty, particularly on the heavy tailed side.



In summary, temperature extremes appear to be relatively insensitive to dynamical downscaling based on comparing
CMIP5 to CORDEX at 0.11° and 0.44°, except over mountains where hot biases decrease with resolution.

### 315  4.1.2 Precipitation extremes

Now we consider precipitation extremes for CMIP5 compared to CORDEX. Figure 3 shows the climatological mean
of Rx1day over the period 1970-2005 for E-OBS and the MMMs of CMIP5 and CORDEX at both resolutions, and
their biases with respect to E-OBS. The left two columns show results for the "common subset" of simulations across
the model categories, and the right two columns for all simulations. The heaviest annual maximum precipitation totals
in E-OBS occur over the Alps and the western side of coastal mountain ranges, including western Norway and north-
eastern Spain. A similar spatial pattern of precipitation distribution can be seen in the models, although totals are lower
in CMIP5, and higher in CORDEX. CMIP5 exhibits a dry bias over most of Europe, particularly over the areas of
maximum precipitation in E-OBS (i.e. over or near mountains), whilst CORDEX exhibits a general wet bias,
particularly in these same locations, and at higher resolution. Results using the entire ensembles are very similar to
using the common subset of simulations. Previous studies suggest that E-OBS underestimates precipitation extremes
since it is not corrected for gauge undercatch and has a relatively low underlying station density (e.g. Prein and Gobiet
2017). Therefore, we also repeat results relative to the MESAN reanalysis (Figure S5) for the shorter period 1989-
2005. MESAN uses a particularly high density of stations in France (see Data section). The climatology of Rx1day is
wetter in MESAN than in E-OBS over most of Europe, most noticeably over the Alps and surrounding areas. This
leads to the dry bias in CMIP5 appearing bigger, and the wet bias in CORDEX decreasing, although it is still present
in the 0.11° simulations. Using regional-scale very high resolution datasets could improve agreement with the 0.11°
simulations, since they tend to give heavier precipitation extremes (Prein and Gobiet 2017). Gauge undercatch could
also contribute to the difference.

Figure S6 shows results for individual models. Again, whilst models agree on the general pattern of precipitation
extremes – i.e. wettest over mountains, there are considerable inter-model differences concerning the magnitude,
particularly over complex orography. A number of CMIP5 models have too light extremes everywhere, but all
underestimate precipitation extremes over mountainous regions to a greater or lesser extent. RCMs systematically
simulate heavier precipitation extremes compared to their driving GCMs, particularly over mountains, and these
extremes tend to become heavier when moving from 0.44° to 0.11° in most cases. Many of the RCMs show a heavy
bias over much of Europe at 0.44°, although this may disappear if compared to MESAN, and this bias gets bigger at
higher resolution and is largest over mountainous regions. Again results are very consistent between ensemble
members of the same models.

Figure 2 (middle column) shows return period curves for Rx1day for Northern, Central and Southern Europe. There is
a clear separation in the tails of the distribution according to resolution, with CMIP5 having the lightest tails, CORDEX
0.44 in the middle, and CORDEX 0.11 with the heaviest tails across all regions (including the subregions – not shown).
Results using the common subset of models or the full ensembles are similar to each other. In order to compare with



observations, E-OBS should be compared to the thin lines for individual models rather than the pooled ensemble
results, since pooling seems to affect the shape of the distribution, causing it to lie below that of the single models.
EOBS tends to lie at the heavy end of the CMIP5 range for southern Europe, between CMIP5 and CORDEX 0.44 for
central Europe, and closer to CORDEX 0.44 in northern Europe. Using MESAN gives slightly heavier tails in central
Europe (figure S7) (particularly in France, where station density is highest –not shown) and more so in southern
Europe, causing the best agreement to occur with CORDEX 0.44 everywhere. Results for Rx5day are similar, but with
marginally less separation between the resolutions, whilst over Northern and Central Europe the best agreement with
E-OBS happens at a slightly higher resolution than for Rx1day – i.e. either with CORDEX 0.44 or the lower end of
the range of CORDEX 0.11 (not shown).

In summary, precipitation extremes are wetter and heavier tailed with higher resolution, especially over mountainous
regions. CMIP5 has a dry bias, particularly over mountains, whilst CORDEX tends to be too wet, particularly at 0.11°,
but results are sensitive to observational dataset used, with wet biases for CORDEX reducing when compared to the
higher resolution MESAN dataset.

### 4.1.3 Wind Extremes

Finally, we examine annual maximum wind (WindXx). Figure 4 shows the multi model means of climatological mean
annual maximum wind over the period 1979-2005 for CMIP5 and CORDEX at 0.44° and 0.11° for the common subset
of simulations and for all simulations compared to three observational datasets. Note however that the model results
are based on the annual maximum of the daily maximum of surface wind (variable "sfcWindmax"), whilst the
observations are based on the annual maximum of 6-hourly data. For CMIP5 models that had both sfcWindmax and
3-hourly data, we compared results using sfcWindmax, 3-hourly and 6-hourly data (Figure S8). 6-hourly data tends to
give lower values than using 3-hourly data or sfcWindmax since some events will be missed due to the lower sampling
frequency. In addition, sfcWindmax does not always appear calculated the same way across all models- although it
often gives higher wind speeds than using 3-hourly or 6-hourly data, the degree to which this is the case seems to
depend on the model, and suggests a different sampling frequency, whilst for a few models sfcWindmax gives the
lowest wind speeds. Ideally an analysis based on a consistent metric across models, such as 3 hourly or 6 hourly wind
speeds, would be performed. However, CORDEX at 0.44° does not have this data available, whilst CORDEX at 0.11°
only has it for a small number of simulations, all of which are based on RCA, and only 3 of which have data for the
driving GCM. Therefore, the reader is invited to interpret results with this caveat in mind. Model sfcWindmax
estimates may also differ in terms of the treatment of surface roughness length and the method for calculating wind at
10m from wind at a higher level.

All three observational datasets are based on ERA-interim, but WFDEI and ECEM-noc are processed differently,
whilst ECEM-wbc is bias corrected using a Weibull distribution based on the station based dataset HadISD (see Data
section). It is notable that all three datasets give different results, particularly ECEM-wbc, despite all being based on
ERA-Interim. WFDEI and ECEM-noc show the same overall pattern of annual maximum wind, with a belt of stronger





winds running zonally across the middle of Europe, including particularly high wind speeds over the British Isles, with
lower wind speeds to the north and south of this band. ECEM-noc has slightly faster wind speeds everywhere. ECEM-
wbc is very different with very high maximum wind speeds over southern Europe and Scandinavia, which are areas
with low wind speeds in the other two datasets. These differences are most apparent for wind extremes- the
climatological means of 6-hourly wind speeds are shown in Figure S9. Although patchier in nature, mean wind in
ECEM-wbc appears broadly similar to the other datasets. However, whilst the other two datasets have similar spatial
patterns between mean and extreme wind, for ECEM-wbc the patterns are very different. One assumption is that the
Weibull correction works well for mean wind, but is not suited to extremes, which are more sensitive to the parameters
of the Weibull correction. It should also be noted that as a reanalysis, ERA-interim is itself model based, albeit with
assimilation of observations.

The CMIP5 driving model mean shows a similar overall pattern of WindXx as WFDEI and ECEM-noc, with a pattern
of weaker winds in the north and south, and a belt of stronger winds in the middle, but with a lower magnitude than
the observations. Using the whole CMIP5 ensemble gives slightly stronger extreme winds. Absolute magnitudes are
not directly comparable to the observational estimates, which would be expected to have slightly slower winds, but
they are nevertheless broadly similar to WFDEI and ECEM-noc, but with too light winds in the central zonal belt. The
CORDEX multi model means show generally higher wind speeds, and a different spatial pattern, with the highest wind
speeds along western coastlines and over mountainous terrain. Differences between the 0.11° and 0.44° runs appear
small. Results for the common subset of simulations are very similar to those obtained from the complete CORDEX
ensembles. Biases are not shown due to the difference in temporal resolution with respect to the observations.

Figure S10 shows that there is a large variety between different models, particularly for CMIP5, but also according to
RCM. CanESM2 and IPSL-CM5A-LR are notable outliers, and this may be related to the timestep of the wind data
used to calculate sfcWindmax in these models. The zonal tripole pattern can be seen in a number of GCMs, as can
stronger winds along western and Mediterranean coastlines, and lower wind speeds over the Alps. Spatial patterns for
the RCMs are very RCM specific and relatively insensitive to driving GCM. All RCMs agree on higher winds over
the British Isles and weaker winds over northern Europe, but notably the mountainous regions have either low or high
wind speeds depending on the model, which must relate to how wind speed is calculated there - it can be imagined that
the wind speed in a valley would be somewhat different to that at the top of a mountain. In terms of differences between
the two resolutions of CORDEX, some RCMs show increased wind speeds with higher resolution e.g. RACMO,
HIRHAM5, and others less so. Again, ensemble members of the same model give similar results.

Figure 2 (right column) shows the return period plots for WindXx for CMIP5 and both resolutions of CORDEX. All
models are shifted to have the same climatology of annual maximum wind for each grid cell, which goes some way to
adjusting for differences in sampling frequency, although there is evidence that the shape of the tails is also affected
for some models (Figure S8). The results for the common subset of CORDEX runs should at least be directly
comparable to each other. The British Isles are shown instead of Northern Europe, since they are particularly affected
by wind extremes, and for comparison with the results for the UPSCALE simulations, where this region shows





0.44 which is in turn heavier than CMIP5, regardless of the subset of models used in creating the pooled ensemble in
almost all regions examined. The CMIP5 results are somewhat sensitive to the models included. The model results
appear relatively consistent with the WFDEI and ECEM-noc observations (note the different sampling frequency).
ECEM-wbc is much heavier tailed in southern and Central Europe.

In summary, winds tend to be stronger, with heavier tails at higher resolution, with a large spread between models.
Observational datasets give very diverse results.

**432   4.2 Global high resolution: UPSCALE**

We now examine the benefits or otherwise of global high vs. standard resolution simulations for simulating climate
extremes. Global high resolution may allow an improved representation of the large scale circulation that cannot be
captured by regional models, which may in turn affect the representation of climate extremes. For this we examine the
UPSCALE simulations (Mizielinski et al. 2014), which consist of a small ensemble of HadGEM3-A simulations at
three different resolutions: 130km (N96), 60km (N216), and 25km (N512) (see Data section).

**438   4.2.1 Temperature extremes**

Figure 5 shows the ensemble mean climatological mean of TXx5day for the UPSCALE simulations over the period
1985-2011 at all three resolutions, and their biases relative to E-OBS. The same general pattern of hotter extremes in
the south and colder in the north and over mountainous regions can be seen at all three resolutions, but temperature
extremes are hotter at higher resolution in the south and east, and colder over mountains. The same pattern of biases is
seen as for CORDEX and CMIP5 with cold biases in the north and hot in the south-east and over mountains. The
mountain biases reduce with higher resolution, as the orography becomes better defined, whilst the hot bias in the SE
and SW increases and the northern cold bias improves slightly. A coastal cold bias at low resolution disappears at
higher resolution, presumably because the ocean influence is carried further over land at low resolution in the large
grid boxes. Note that the SSTs are prescribed and are the same for all simulations. Results for TXx are similar but
hotter (not shown).

Figure 6 (left column) shows regional return period plots for TXx5day for the UPSCALE simulations. Results are a
little less consistent across regions for UPSCALE compared to the CMIP5 vs CORDEX analysis, so we split Northern
Europe into the British Isles and Scandinavia, and add the Alps, to better capture regional variations. Again, the thin
lines are individual simulations, and the circles are for results pooled across the ensemble members for each resolution
separately. Since the pooled ensembles are only based on one model, results are presented without adjusting according
to the climatology of TXx5day, although bias adjusted results can be seen in Figure S11 and allow differences in the
shapes of the tails to be seen more clearly. TXx5day seems to be hotter with higher resolution over most regions, with
the notable exception of the Alps, where the higher elevations with higher resolution give rise to colder temperature





extremes. There are notable biases relative to the observations, with the models being too cold in the north, especially
at low resolution, whilst in the south the colder subset of models (N96, the lowest UPSCALE resolution) agree with
the observations. Over the Alps, again the low resolution simulations agree best with observations, with the warmest
temperatures, but this will depend on the height of the meteorological stations. This apparent contradiction to the
reduced orographic hot bias with resolution in figure 5 comes from the stronger cold bias of the surrounding areas at
low resolution. Figure S11 shows that in general there is not much difference between the shape of the tails with
resolution, with only slightly heavier tails with increased resolution over the British Isles and Southern Europe.
Agreement with E-OBS is very good everywhere. Results for TXx are similar.

In summary, hot biases of temperature extremes over mountains reduce with increased resolution for HadGEM3-A.
Elsewhere extremes get hotter with resolution, whilst the shapes of the statistical distributions are insensitive.

### 4.2.2 Precipitation extremes

For precipitation, Figure 7 shows the ensemble mean climatological mean of Rx1day for the period 1989-2010 for the
three UPSCALE ensembles and their biases relative to E-OBS. The overall pattern of Rx1day is similar to that in E-
OBS, with heaver precipitation extremes and finer spatial detail with increasing resolution over complex orography.
The N96 runs have an area of heavy precipitation stretching from France into Germany, whilst the N216 and N512
simulations show instead a pattern of heavy precipitation either side of the Alps, with a drier area in-between. A general
wet bias can be seen at all resolutions over Europe, whilst the dry bias over orography in the Alps, Southern Norway
and Scottish Highlands reduces with resolution and a wet bias on the southern edge of the Alps and the coastal side of
the Dinarie Alps in the Balkans appears as resolution increases. Comparing to MESAN instead of E-OBS, the general
wet bias disappears, and the dry mountain bias over orography at low resolution increases. The differences between
resolutions appear smaller than for the CMIP5 versus CORDEX analysis: all the UPSCALE simulations look most
similar to CORDEX at 0.44°. However, UPSCALE does not reach as fine a resolution as CORDEX at 0.11° (25 km
vs 12.5 km), and CMIP5 is on average slightly coarser than the N96 simulations. In addition, it should be noted that a
model's nominal resolution does not always accurately reflect the spatial scales that it can represent. Results are similar
for Rx5day (not shown).

Figure 6 (middle column) shows the return period plots for Rx1day for the three resolutions of UPSCALE ensembles.
Slightly heavier precipitation extremes are found at higher resolution in all the regions shown (exceptions are France
and Mid Europe- not shown), although differences are small, they are more obvious in southern Europe and especially
in the Alps. Figure S11 shows that there is not much difference in the shape of the tails for most regions, although
there are very slightly heavier tails at higher resolution for southern Europe and more obvious differences over the
Alps in the same direction, both of which are regions where convective precipitation is important. E-OBS tends to lie
just below the model simulations for most regions (Figure 6 – compare with the thin coloured lines), although it agrees
with the models for the British Isles, and is between the low and medium resolution simulations over the Alps. MESAN
gives higher values for observed Rx1day which improves agreement in regions where E-OBS lay below the models,





and causes a higher resolution subset to agree better in the other regions (Figure 6). For the bias adjusted versions E-
OBS tends to lie just on the lower end of the ensemble for most regions, whilst MESAN gives slightly heavier tails
and tends to improve agreement with models (Figure S11). Results for Rx5day are broadly similar (except that both
sets of observations lie above all the models for the British Isles).

In summary, precipitation extremes are somewhat wetter and heavier tailed with increasing resolution mostly in
southern Europe and the Alps for HadGEM3-A. Dry orographic biases decrease with resolution but wet biases appear
in the south next to mountain ranges instead.
**4.2.3 Wind extremes**
For wind extremes, Figure 8 shows the spatial patterns of climatological mean annual maximum wind based on 6-
hourly data for UPSCALE and the same for three observational datasets. In this case the models and observations are
directly comparable since they share the same temporal resolution. The spatial patterns are similar for the three
different model resolutions, with the highest winds over the British Isles and coastal regions, lower wind speeds over
the Alps, and the zonal tripole pattern described above, although this does not extend as far east as in the observations
(i.e. ECEM-noc and WFDEI). The main differences are that the lower resolution model (N96) has stronger winds
around the British Isles and western coastlines, presumably because the larger grid boxes overlap more with the ocean,
which tends to have higher wind speeds. The wind speeds at higher resolution are a little stronger overall, most
obviously in the central European zonal belt, and over the Alps and Norwegian mountains. As noted before, the
observational estimates vary significantly and therefore the biases depend on the observational dataset used, i.e.
extreme winds are slightly weak compared to ECEM-noc over much of Europe; compared to WFDEI, winds are too
strong in the north and south, and too weak in the east; and compared to ECEM-wbc, winds are far too weak in the
north and south and too strong in-between.

Figure 6 (right column) shows the return period plots for annual maximum wind for the UPSCALE simulations,
without shifting the climatology. Over most regions the strongest extreme winds are found at the highest resolution,
with the exception of the British Isles (and the Iberian Peninsula- not shown) where the low resolution models have
the strongest winds. This is likely related to the large coastal grid boxes overlapping windy ocean areas as discussed
above. As noted above, there are large differences between observational estimates, with ECEM-wbc having
considerably higher values and heavier tails than the other two datasets and models over most regions, except the
British Isles. ECEM-noc tends to agree best with the model simulations, whilst WFDEI tends to lie at the lower end of
the model range or underneath. For the bias adjusted versions of the return period plots (Figure S11), differences in
the shapes of the tails with resolution are generally small, although with marginally heavier tails with increasing
resolution over a number of regions. Agreement of the shape of the tails with ECEM-noc and WFDEI is good.

In summary winds are slightly stronger and heavier tailed at higher resolution in HadGEM3-A, except over coastal
areas where large grid boxes at low resolution bring strong ocean winds further over land.





### 4.3 Circulation Analogues


For the global model results, any differences in the representation of extremes according to resolution could come from
either upscaling or downscaling effects. Upscaling effects could include a better representation of the large scale
circulation, whilst downscaling allows a better representation of small scale processes, such as convection, and an
improved representation of orography and coastlines. In order to investigate which of these effects leads to the
differences between the low (N96) and high resolution (N512) HadGEM3-A simulations, we employ a circulation
analogue technique (e.g. Vautard et al., 2016), which is frequently used in attribution studies (see e.g. Stott et al., 2016;
Cattiaux et al., 2010). The idea is to determine whether the simulation of climate extremes changes between the two
resolutions if both were to have the same large scale circulation –i.e. isolating the downscaling effect, or conversely
whether circulation differences explain any differences in extreme events whilst circulation-variable (e.g. precipitation)
relationships stay the same –i.e. the upscaling effect.

For each day in the lower resolution simulations we pick the nearest circulation analogue from anywhere in the higher
resolution simulations, providing it happens at the right time of year (i.e. within a 30-day window centred on the day
of the year in question). We then record the associated temperature, precipitation and wind values from the higher
resolution simulations to make a "$u$-chronic" dataset (e.g. Jézéquel, et al. 2018) that contains data from the high
resolution simulations but follows the daily sequence of circulation patterns from the low resolution models. We then
repeat the analysis of return periods and value as above. We also do the reverse (find analogues for the N512 circulation
in the N96 ensemble and record the N96 temperature). Since results using analogues are not directly comparable to the
original results, due to lack of exact analogue match, we also perform "self-analogues" -i.e. finding circulation
analogues for the N96 simulations within the N96 ensemble, (excluding the same year from the same ensemble
member) and creating a u-chronic time series, and the same for the N512 ensemble). Comparing the resulting return
period curves tells us about the contribution of large-scale circulation and downscaling to differences in extremes
between the two resolutions. For example, comparing the N96 self-analogue return curve to the version based on N512
circulation but with N96 precipitation shows us the contribution of any differences in the large scale circulation
between the resolutions i.e. the upscaling effect. Comparing the N96 self-analogue to the version based on N96
circulation with N512 precipitation shows us the downscaling effect – i.e. any difference between the relationship
between the large scale circulation and precipitation.

Analogues are defined using geopotential height at 500 hpa, since this avoids complications relating to surface heat
lows associated with heat waves in anticyclonic conditions that occur in summer, whilst also avoiding incomplete data
due to mountain ranges. Geopotential height is regridded to a 2° grid using bilinear interpolation. This choice ensures
that we are comparing analogues with the same resolution and do not penalise small-scale differences. Similarity
between circulation states is calculated using the Euclidean distance. For precipitation and wind the European domain
used is -16 to 44° E and 34 to 72° N (roughly the same as the domain plotted in the map-based figures). For temperature,
a larger domain is used, since the history and trajectory of air masses is important for temperature extremes. This
domain is loosely based on the domain used by Cattiaux et al. (2010) and extends over the N. Atlantic as well as





Europe, (-62 to 44°E and 24 to 80° N). However, results are very similar if the smaller domain is used (not shown).
For the 5-day variables (Rx5day and TXx5day); daily geopotential height, precipitation and temperature datasets were
smoothed using a 5-day running mean first, and then analogues were calculated, and the u-chronic datasets constructed.
We also tried doing the 5-day means last rather than first, i.e. calculating analogues using daily data and smoothing
the u-chronic dataset. The relationship between the different curves was largely consistent between the two techniques,
but absolute values differed and the shape of the distributions changed a little. Results presented here are based on the
first technique since it replicates better the autocorrelation structure of the original analysis.

Figure 9 shows the results of the analogue analysis. The blue curves show the results for the N512 self-analogues, grey
represents the N96 self-analogues, red represents results using the circulation patterns from the N96 runs but with the
N512 circulation-variable relationships, and green indicates N512 circulation with N96 circulation-variable
relationships. The difference between the blue and red curves (or the grey and green curves) shows the contribution
from differences in the large scale circulation with resolution, whilst the difference between the blue and green curves
(or the red and grey curves) indicates the downscaling effect.

For TXx5day downscaling effects are dominant over regions that have a clear difference between resolutions, although
circulation differences also have a small effect in some regions (Figure 9). For Rx1day the different curves are very
close together for most regions, making it difficult to discern the relative contributions from upscaling and
downscaling. However, it generally seems to be downscaling effects that are the most important, and this can be seen
more clearly for the Alps and Southern Europe where there are larger differences with resolution. Interestingly, these
are regions where convective precipitation is particularly important for precipitation extremes.  For wind extremes
downscaling effects also dominate, but the large scale circulation also plays a role in Scandinavia. Results for TXx and
Rx5day are very similar to those for TXx5day and Rx1day respectively (not shown).

Also shown, using thick solid lines, are the original pooled ensemble results without using analogues. By comparing
these with the self-analogue results (i.e. compare the blue line with the blue circles for N512, and the grey line with
the grey circles for N96), we can see how successful the analogue technique is in recreating the original distributions.
The self-analogue results tend to be close to, but slightly below the original results for wind and Rx1day, with a slight
difference in the shape of the tail at the far right for Rx1day. This can be explained by the fact that the analogues are
not perfect, and since the circulation patterns associated with climate extremes are rare, the nearest analogues are likely
to represent slightly less severe events. The original results are beneath the analogue results and a different shape for
TXx5day, which seems to be associated with the 5-day averaging, and is much less marked for TXx (not shown).
Undertaking the 5-day averaging last rather than first (see Methods) shifts analogue results downwards, underneath
the original curves, but otherwise gives the same results (not shown). The same phenomenon is seen for Rx5day (not
shown).

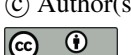



In summary, for all three types of extreme events, downscaling effects appear to dominate the differences seen between
the 130km and 25km HadGEM3-A simulations. This suggests that at least for this model, any large scale circulation
differences obtained with global high resolution do not affect the statistics of these extreme events much.

## 5 Summary and Discussion

### 5.1 Summary

We evaluated climate model simulations of temperature, precipitation and wind extremes over Europe, addressing
three questions: 1) The benefits of dynamical downscaling using regional climate models by comparing EURO-
CORDEX simulations at two resolutions (12.5 and 50 km) to their driving coarser resolution CMIP5 models; 2) The
benefits of increased resolution for global models by comparing HadGEM3-A simulations at three resolutions (130,
60 and 25 km); and 3) whether any differences according to resolution in the global model comes from differences in
the large scale circulation (upscaling) or the representation of small scale processes, and features (downscaling) using
a circulation analogue method.

For temperature extremes, increased resolution did not make much difference to results for the CORDEX vs CMIP5
analysis, both in terms of the shapes of the distributions, which all agreed well with observations, or in terms of biases,
apart from reducing hot biases over mountains. This reduction in orographic bias with increased resolution was also
seen in the HadGEM3-A GCM simulations, along with a general increase in magnitude of hot extremes elsewhere,
which reduces biases in the north, and increases them in the south. Overall the benefits of increasing resolution were
limited, or region dependent.

Precipitation extremes were more sensitive to resolution, particularly in the CMIP5 vs CORDEX analysis, with heavier
tails at higher resolution across all regions. Spatially, CMIP5 shows a general dry bias compared to E-OBS, particularly
over mountainous regions, whilst CORDEX shows the opposite, with increasing wet biases at 0.11° compared to 0.44°,
which appears to be systematic across models. The higher resolution MESAN reanalysis gave wetter extremes and
heavier tails than E-OBS, agreeing best with the 0.44° resolution CORDEX simulations, highlighting the importance
of the choice of observational dataset. Differences according to resolution were smaller for the global scale HadGEM3-
A simulations, although these span a smaller range of resolutions. Differences were most obvious in southern regions
and the Alps, with heavier tails and wetter extremes at higher resolution. Dry biases over orography decreased with
increasing resolution; however, wet biases next to some mountain ranges in the south emerge. Return period curves
for HadGEM3-A tended to agree well with MESAN, but were too wet compared to E-OBS.

For wind extremes, higher resolution gave stronger winds and heavier tails for most regions for both the CORDEX vs
CMIP5 analysis and to a lesser extent for HadGEM3-A. The largest differences were between CMIP5 and CORDEX
at 0.44°, with less difference between the two resolutions of CORDEX. Differences between observational estimates



made model evaluation difficult, whilst inconsistencies in the way daily maximum wind is calculated in different
models were also an issue.

The circulation analogue analysis suggested that for the global scale HadGEM3-A simulations, differences according
to resolution for all three phenomena were dominated by downscaling effects, with only small contributions from
differences in the large-scale circulation.

**5.2 Discussion**
For temperature extremes, our results imply that increased resolution in both regional and global models is of limited
benefit at the resolution range considered here, except in reducing the hot bias over mountainous areas. In particular,
for resolutions used in the UPSCALE experiments, we do not find strong upscaling nor downscaling effects. These
findings agree with Vautard et al. (2013) for regional models, who find limited benefits in simulating various aspects
of heatwaves between the 0.44° and 0.11° versions of the EURO-CORDEX models. However, our results for the
global model analysis are based on only one model and the new model simulations and analyses being generated as
part of the PRIMAVERA and HighResMIP projects (https://www.primavera-h2020.eu/; Roberts et al. 2018; Haarsma
et al. 2016) will be very useful for determining how representative our results for HadGEM3-A are of other GCMs.
For instance, improvements in the simulation of summer blocking, which can be involved in heatwave generation is
very model dependent (Scheimann et al. 2014). Furthermore, Cattiaux et al. (2013) find that the frequency, intensity
and duration of summer heatwaves improve in the IPSL model with resolution, associated with a better representation
of the large scale circulation. In addition, here we examine only one aspect of heat waves (intensity), and it could be
that results are different for others aspects, such as frequency, duration and timing.

For precipitation extremes, we found that the CMIP5 models were too dry whilst CORDEX was a little too wet at
0.44° and more so at 0.11° when compared to E-OBS. This was particularly the case over complex orography. This is
consistent with results for mean precipitation in EURO-CORDEX in Kotlarski et al. (2014). However, our results
depend on the observational dataset compared against, with MESAN giving heavier extremes than E-OBS and agreeing
reasonably well with the 0.44° simulations. Other studies suggest that country-scale higher resolution precipitation
datasets give heavier precipitation extremes still, which may agree best with the 0.11° simulations. Similarly, for mean
precipitation, Prein and Gobeit (2017) find that RCM biases are a similar size to the differences between different
observational estimates. For extreme precipitation, Prein et al (2016) and Torma et al (2015) find that various aspects
(biases, frequency-intensity distributions, spatial patterns) of mean and extreme precipitation improve in EURO-
CORDEX at 0.11° compared to 0.44° when compared to such datasets for Europe and the Alps respectively. Prein et
al (2016) ascribe this mostly to the better representation of orography at higher resolution, but also the ability to capture
the larger scales of convection. However, some of the difference in our results may also be explained by
parameterisation schemes that tend to be tuned to one resolution and can behave sub-optimally at others.





For the UPSCALE global simulations, there was less difference with resolution, with the biggest differences over or
near mountains. However, these simulations span a narrower range of resolutions, i.e. not reaching the same high
resolutions as CORDEX 0.11°, but also not as coarse as some CMIP5 models. Other global model studies also tend to
find an increase in precipitation extremes with increased resolution for Europe, which is continent-wide in summer,
and concentrated in mountainous regions in winter (Volosciuk et al. 2015; Wehner et al. 2014). This sometimes
improves agreement with observations (e.g. Kopparla et al. 2013; Wehner et al. 2014 for winter), but can overestimates
summer extreme precipitation if parameterisation schemes are not retuned (Wehner et al. 2014).

For wind extremes, our findings of stronger winds and heavier tails with increased resolution are consistent with
previous studies (e.g. Pryor et al. 2012; Champion et al. 2011; Kunz et al. 2010). However, observational issues made
model evaluation difficult.

The results of the circulation analogue analysis on the HadGEM3-A GCM simulations suggested that downscaling
effects were the dominant cause of differences with resolution for all three phenomena, with limited effects of any
differences in the representation of the large scale circulation. If this result also applied to other GCMs, it would suggest
that dynamical downscaling with more economical limited area models would be a better strategy for simulating
European extreme events, whilst GCM efforts could focus on other aspects such as multiple members or multi-physics
ensembles. However, we cannot reach this conclusion based solely on this analysis, since we examine only a single
model, which may not be representative of other models, and because the range of resolutions considered may be too
narrow. Furthermore, a number of studies do find improvements in the large-scale circulation with resolution, including
for extra-tropical cyclones and storm tracks (Colle et al. 2013; Jung et al 2006; 2012, Zappa et al. 2013), Euro-Atlantic
weather regimes (Dawson et al. 2012; 2015; Cattiaux et al. 2013) and blocking (Jung et al. 2012, Anstey et al. 2013;
Matsueda et al. 2009, Berckmans et al 2013; Scheimann et al. 2014; Davini et al 2017a; 2017b; see also Introduction).
Interestingly, Scheimann et al. (2017) find improvements in Euro-Atlantic blocking with resolution in all seasons in
the same HadGEM3-A simulations as we analyse here. However, the net effects on extremes, given all uncertainties,
was not explicitly investigated. Our study does not seem to be able to discern such effects.

Overall our results suggest that whether or not increased resolution is beneficial for the simulation of extreme events
over Europe depends on the event being considered. Benefits appear limited for heatwaves, whereas wind extremes
and particularly precipitation extremes are more sensitive. We do not find any particular advantage in using a global
high resolution model compared to regional dynamical downscaling, with the caveats that this investigation needs to
be extended to other GCMs, and a wider range of resolutions should be investigated.

In order to fully address the question of the benefits of increased resolution for European climate extremes, a number
of aspects remain to be investigated. Firstly, the analysis could be widened to other types of extremes, for example,
sea level rise and storm surge, or other aspects of extremes could be considered e.g. timing, frequency and duration of
events. The global simulations we investigated were atmosphere-only, and the role of increased ocean resolution and
also vertical resolution and model top height should be considered. Finally, we assume that better historical





performance translates into more accurate future projections. Lhotka et al. (2018) find low sensitivity of heatwave
projections to resolution in EURO-CORDEX RCMs. However, Van Haren et al. (2015b) find stronger future summer
drying and heating in central Europe with increased resolution in the EC-Earth GCM due to differences in atmospheric
circulation. Concerning precipitation, future projections for large scale and seasonal mean precipitation are consistent
between large scale and convective permitting models, whereas summer daily and sub-daily intensities increase more
in the future in convection permitting models (Kendon et al. 2017; Ban et al. 2015; Kendon et al. 2014). For wind,
Willison et al. (2015) find a larger response of the North Atlantic storm track to global warming with higher resolution
in the WRF model. The sensitivity of projections to resolution nevertheless remains an area that needs further research.
**Data and code availability**
The CMIP5 and CORDEX data used for this analysis are available from the Earth System Grid Federation portals, and
are detailed in Table S1. The HadGEM3-A UPSCALE simulations are available from the CEDA-JASMIN platform.
E-OBS can be downloaded here https://www.ecad.eu/download/ensembles/download.php, MESAN is available here
http://exporter.nsc.liu.se/620eed0cb2c74c859f7d6db81742e114/, access to WFDEI is detailed here http://www.eu-
watch.org/gfx_content/documents/README-WFDEI%20(v2016).pdf and ECEM data are available from the
Copernicus Climate Data Store https://cds.climate.copernicus.eu.
**Author contributions**
CI, RV and SJ conceptualised the study, CI carried out the analysis and wrote the manuscript, JS managed the CRECP
project together with CH and BE, and all co-authors were involved in discussions to prepare the study and helped
improve the manuscript.
**Competing interests**
The authors declare that they have no conflict of interest.
**Acknowledgements**
This work is published in the name of the European Commission, with funding from the European Union through the
Copernicus Climate Change Service project C3S_34a Lot 3 (Copernicus Roadmap for European Climate Projections).
The Commission is not responsible for any use that many be made of the information contained. We acknowledge the
WCRP's Working Group on Regional Climate, and the Working Group on Coupled Modelling - the coordinating body
of CORDEX and the panel responsible for CMIP5 respectively. We thank the climate modelling groups for producing
and making available the model output listed in Supplementary Table 1, which is available at http://pcmdi9.llnl.gov.
For CMIP, the US Department of Energy's Program for Climate Model Diagnosis and Intercomparison provides
coordinating support and led development of software infrastructure in partnership with the Global Organization for
Earth System Science Portals. We thank the modelling team that produced the UPSCALE simulations, and



acknowlegde the JASMIN and IPSL mesocentre computing clusters on which this analysis was performed. We also
acknowledge helpful input from the CRECP project scientific advisory board and useful discussions with UK Met
Office Scientists, in particular Malcolm Roberts and Carol McSweeney.

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



**Figures**



**Figure 1: Climatological mean of TXx5day for the period 1970-2005 for (a) EOBS, the multi model mean of the common subset of models for (b) CMIP5, (f) CORDEX 0.44° and (j) CORDEX 0.11°, (c, g, k) their biases with respect to EOBS, and (d,e,h,i,j,k) the same for the full ensembles of CMIP5, and CORDEX. Units °C.**








**Figure 2: Return period plots for (left) TXx5day, middle column Rx1day and (right) annual maximum wind, for CMIP5**
**and CORDEX for Northern Europe (top row except top left = British Isles), Central Europe (middle row) and Southern**
**Europe (bottom row). CMIP5 is shown in grey, CORDEX 0.44° in red and CORDEX 0.11° in blue. Thin lines are individual**
**ensemble members, circles represent the pooled ensembles, lighter shades for the full ensembles, and darker shades for the**
**subset of models common to CMIP5, and both CORDEX resolutions. Observations are shown in black, circles for E-OBS**
**temperature and precipitation and WFDEI wind, triangles for MESAN precipitation and ECEM noc wind and crosses for**
**ECEM wbc wind. Confidence intervals based on bootstrapping are shown with dashed lines. The time periods considered**
**are 1970-2005 for TXx5day and Rx1day, and 1979-2005 for wind.**



Figure 3: As for Figure 1 but for the climatological mean of Rx1day. Units mm.

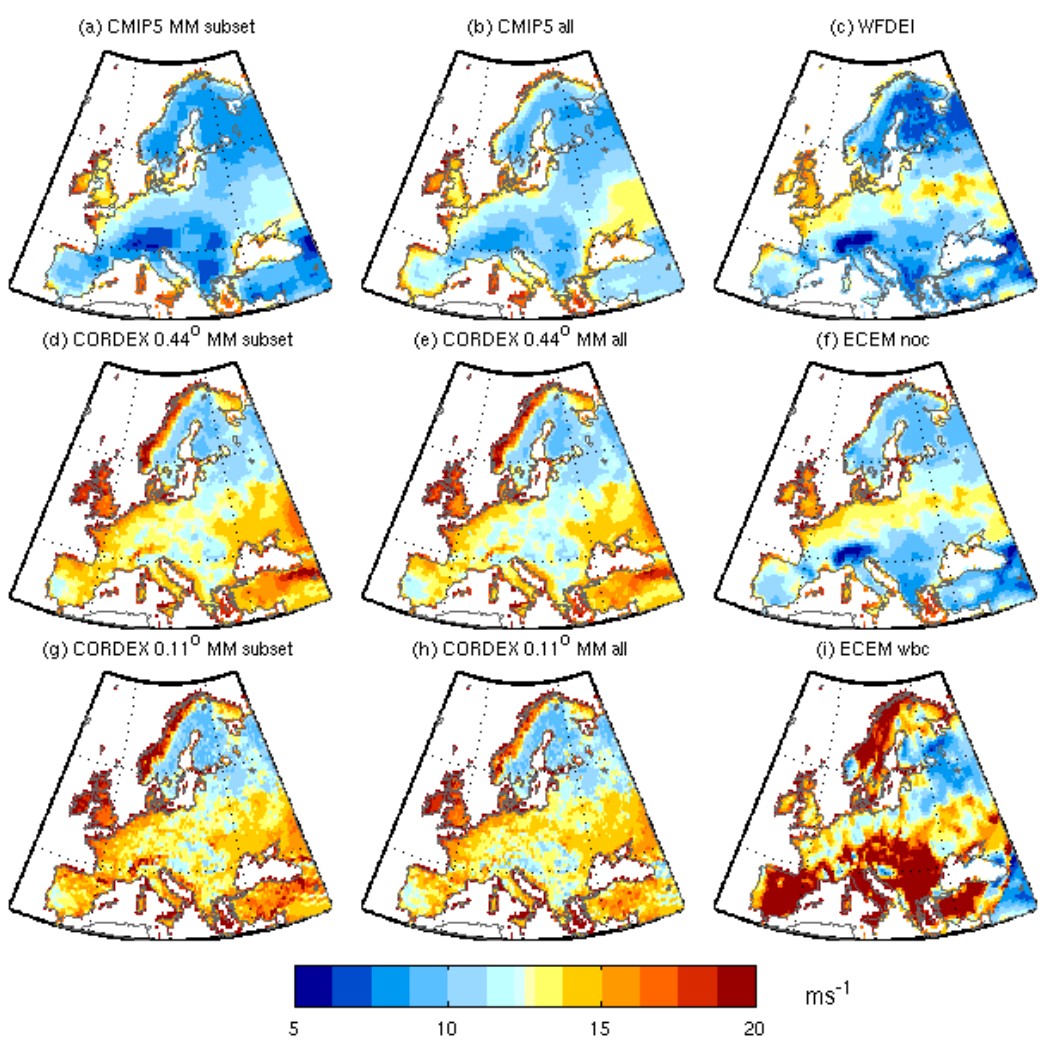



**Figure 4: Climatological mean of annual maximum of daily maximum wind for the period 1979-2005 for the multi model mean of the common subset of models for (a) CMIP5, (d) CORDEX 0.44° and (g) CORDEX 0.11°, (b, e, h) the same for the full ensembles of CMIP5 and CORDEX, and the observational datasets (c) WFDEI, (f) ECEM noc (i) ECEM wbc. Units meters per second.**



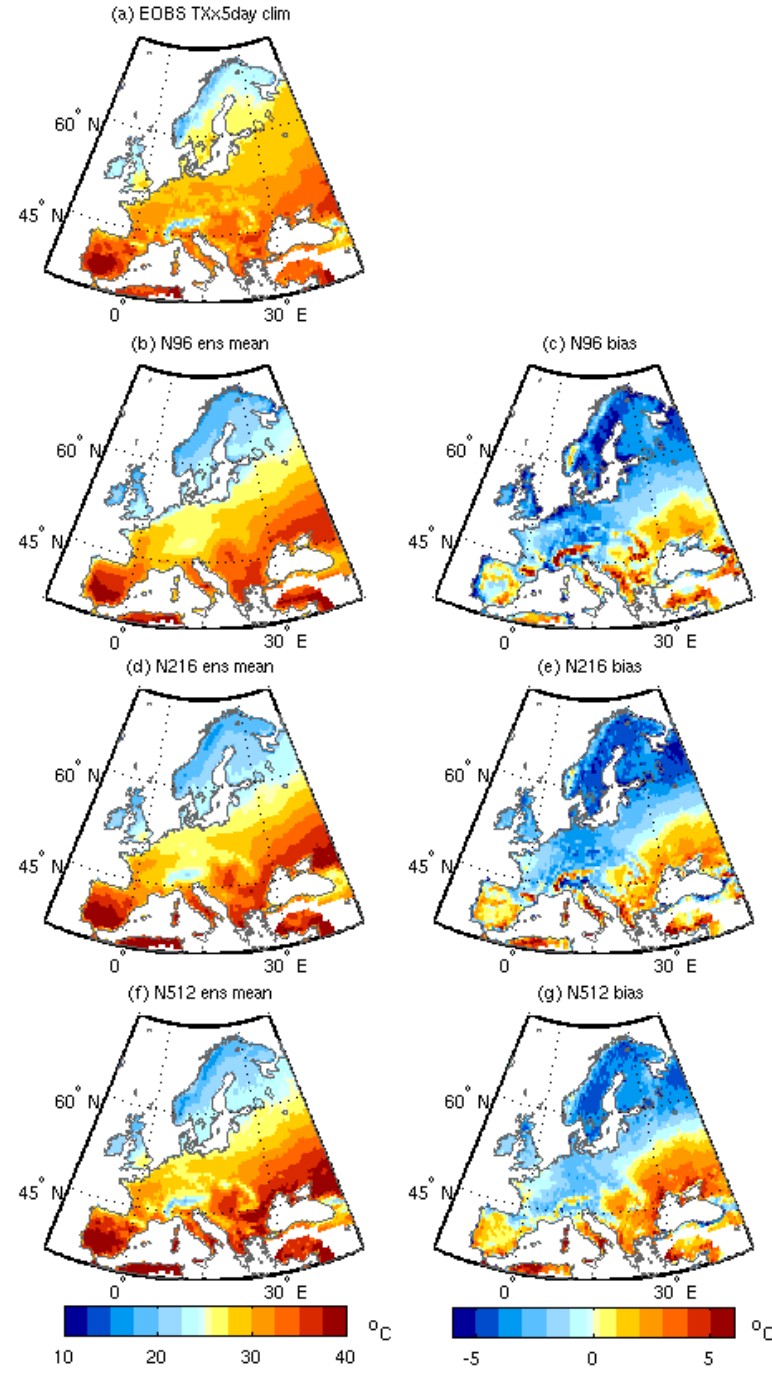

1065 **Figure 5: Climatological mean of TXx5day for the ensemble means of three resolutions of HadGEM3-A (UPSCALE) GCM simulations (left) for the period 1985-2011 and their biases with respect to E-OBS (right). (a) EOBS, (b, c) N96 (130 km), (d, e) N216 (60 km), (f, g) N512 (25 km). Units °C.**





**Figure 6: Return period plots for (left) TXx5day, middle column Rx1day and (right) annual maximum wind, for the UPSCALE simulations for (top row) the British Isles, (2ⁿᵈ row) Scandinavia, (3ʳᵈ row) Central Europe, (4ᵗʰ row) Southern Europe, and (last row) the Alps. N96 is shown in grey, N216 in red and N512 in blue. Thin lines are individual ensemble members, circles represent**



the pooled ensembles. Observations are shown in black, circles for E-OBS and WFDEI, triangles for MESAN and ECEM noc, and asterisks for ECEM wbc. Confidence intervals based on bootstrapping are shown with dashed lines. The time periods considered are 1985-2011 for TXx5day, 1989-2010 for Rx1day, and 1986-2011 for wind. NB: there is no bias correction of the climatology (see methods).

**Figure 7: Climatological mean of Rx1day for the ensemble means of three resolutions of UPSCALE (left) simulations for the period 1989-2010 and their biases with respect to E-OBS (middle) and the MESAN reanalysis (right). (a) EOBS, (b) MESAN (c-e) N96, (f-h) N216, (i-k) N512. Units mm.**

Figure 8: Climatological mean of annual maximum wind for the ensemble means of three resolutions of UPSCALE (left) simulations for the period 1986-2011 and their biases with respect to the observational datasets ECEM wbc (left), ECEM noc (middle) and WFDEI (right). (a) ECEM wbc, (b) ECEM noc (c) WFDEI, (d-g) N96, (h-k) N216, (l-o) N512. Units meters per second.

1080

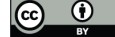







1085

**Figure 9: Circulation analogue results. Return period plots for (left) TXx5day, (middle) Rx1day and (right) annual maximum wind for (top) the British Isles, (2nd row) Scandinavia, (3rd row) Central Europe, (4th row) Southern Europe and (5th row) the Alps. Grey represents the N96 self-analogues, blue the N512 self-analogues, red is for N96 circulation with N512 variables (e.g. precipitation) and green is for N512 circulation with N96 variables. Thin lines represent individual ensemble members, circles represent results pooled across ensemble members. Dashed lines are 5-95% confidence intervals based on a bootstrapping technique. Thick blue line represents the original pooled N512 results like those shown in Figure 6 (although sometimes based on a different time period), and**

1090

**the thick grey line represents the equivalent for the N96 simulations. Results for TXx5day are based on the period 1985-2011, Rx1day 1986-2011, and wind 1986-2011.**