# Peer review of "The benefits of increasing resolution in global and regional climate simulations for European climate extremes"

_Geoscientific Model Development, 2019_

## Referee Comment (RC1) · Anonymous Referee #1 · 25 Nov 2019

The paper, in general, is well written and the authors attempt to solve a very critical issue regarding the added value of increasing resolution. However I do have some issues regarding the dataset used and some of the methodologies applied, hence I recommend major revision.

Here my major comments:

a) All the datasets used to compare the models' output are labelled as observations when, in reality, they are not, they are reanalyses. So I wonder why the authors chose a reanalysis when there are real observational datasets, like GPCP, CHIRP or TRMM. I do know all those datasets have some caveats, but they are derived from direct obser-

vations, (and most of them are of the same resolution like the one that the reanalysis used in the paper). At least for precipitation and temperature, I suggest that the authors re-do the comparisons with some of the observational datasets mentioned. For wind, I understand that there might not be another option that the one that they used.

b) I do not understand why the authors chose to use climatological means of Txxand Rx1day. Would not be more useful to have seasonal maps? Mostly for precipitation, as extremes, can occur either in summer or in winter but by very different processes. In this way, we could have seen which type of seasonal extremes are better (or worse) capture and this might also complement the results of the analogues.

c) On the same idea, why to use a 5-day average (line 568) mainly for precipitation, given the fact that by doing that you are smoothing the extreme event that normally would last one day. For temperature, it might not be a problem, considering that heat-waves per definition last several days(at least 3).

General comments:

1. Line 40, please define what do you mean by "small scale", do you mean synoptic, mesoscale?

2. Line 60, remove "really"

3. Line 62, added to the list of reference: Risanto, C.B.; Castro, C.L.; Moker, J.M., Jr.; Arellano, A.F., Jr.; Adams, D.K.; Fierro, L.M.; Minjarez Sosa, C.M. Evaluating Forecast Skills of Moisture from Convective-Permitting WRF-ARW Model during 2017 North American Monsoon Season. Atmosphere 2019, 10, 694.

4. Line 66-67, I think the justification of centre this paper on Europe, has to other than" "its climate is highly variable and affected by a range of both large and small scale processes which present challenges for adequate simulation", as this is true for several regions in the planet.

5. Line 56: " . . .history and trajectory of air masses ARE important . . . (instead of"is")

---

## Referee Comment (RC2) · Anonymous Referee #2 · 1 Dec 2019

In the manuscript "The benefits of increasing resolution in global and regional climate simulations for European climate extremes" Iles et al. assess the dependence of simulating extreme event intensity on model resolution and model strategy (regional vs. global) over Europe. They show that higher resolution simulations have generally stronger extremes with higher sensitivities for precipitation and wind extremes than for temperature extremes. Their results generally confirm previous studies and for me, the greatest value of this paper is the combination of a large range of models and investigating three extreme events, which provides a good overview. I also like the upscaling analysis although it is kept fairly brief. However, I have concerns in how the team compared models with different resolutions and would like to see a more careful

usage of the word model bias especially in the context of precipitation and wind analysis for which the used observational datasets are of questionable quality. Below is a more detailed description of my major/general and minor comments.

Major/general comments:

1. I have concerns with using nearest neighbor remapping for extreme value analysis. This remapping method will artificially increase extremes in high-resolution data particularly in situations with strong gradients. For instance, precipitation extremes are typically very localized and have strong spatial gradients. You are remapping the 0.11 EURO-CORDEX simulations to a 0.5deg grid. This means that you pick the closest 0.11deg grid cell to each 0.5deg cell and assume that it rains ass much on the 0.5deg area than on the 0.11deg area. This is certainly not the case and by doing this, you violate mass conservation of precipitation. I strongly recommend using a conservative remapping method that conserves mass and energy while remapping. This will affect many of your results since you generally find that extremes are more intense in the high-resolution simulations. Using conservative remapping will dampen this effect and might change your conclusions.

2. Along coastlines, you have to be careful that you only consider land grid points in your evaluation against land-based observations. As you mention in your wind speed analysis, there is a sharp gradient in wind speed but also for temperature extremes between ocean and land. I am not sure if your interpretation that high wind extremes propagate further inland in coarse-scale models is correct. Could it be that you include ocean grid cells in your analysis, which cause a high bias in low-resolution models?

3. Please use a lapse rate adjustment when comparing temperature extremes between model grid spacings. Your finding that heat extremes are less overestimated in high-resolution simulations in topographic regions is trivial since the coarse resolution models have lower topography and therefore higher temperatures. You could simply use the topography of E-OBS and correct the model temperature with a climatological

average lapse rate (e.g., 6.5 deg C per km).

4. Please avoid writing about model biases when you compare extreme precipitation to E-OBS. You mention that E-OBS underestimates precipitation extremes, which means that you would like a model to be wetter than E-OBS. Evan the MESAN precipitation extremes are likely too low. I suggest being very careful how you use the word bias and rather use difference when comparing to precipitation but also wind datasets.

5. The separation of CMIP5 and CORDEX results from the UPSCALE results make the paper unnecessary long and harder to read. I recommend combining sections 4.1 and 4.2 since I do not see a good reason to separate the UPSCALE ensemble from the other datasets. Also, combining figures 1, 3, and 4 with 5, 7, and 8 would be beneficial. You could do this by not showing the results for all and common ensemble members but only one of the two. They are very similar anyway.

6. Also, the summary and discussion section could be combined. After the summary section you again briefly summarize results in the discussion. You would lose very little information by removing the summary section.

7. I am concerned with how different the three wind observations are. Are they all equally likely? Looking at the big differences between these observations makes me wonder if you can/should evaluate the models at all. Just looking at your fields in Fig. 4 (c,f,i) makes me wonder if these datasets can even capture topographic effects. The low wind extreme minimum over the Alps looks very unrealistic and event he models seem to do a better job capturing these effects that the observations. Would the use of MESAN winds be a better option?

8. Please be careful when you use the term model resolution and do not confuse it with model horizontal grid spacing. The model resolution is typically 4 to 8 times the model horizontal grid spacing (Skamarock 2014)

Minor Comments:

L37-8: Please explain what a long-standing anticyclone is? Do you mean a stationary anticyclone?

L41: I would say "flash floods" here since river floods involve large-scale processes.

L43: I suggest "poorly resolved" since some of these processes can be resolved at fairly large grid spacings.

L50: Could you please be more specific than "small-scale processes and features involved"

L89: What do you mean with "their" in "to their information"?

L136: E-OBS has quite high station density in some regions (e.g., Slovenia, Germany) but low density in others (e.g., Austria, Spain)

L168L: "us to identify"

L173: Are these daily maximum surface wind speeds based on model time step wind speed or hourly maxima of instantaneous wind, or something else?

L174: "Those consist of"

L192: What is this alternative dataset for SST?

L203: Performing your analysis on a 0.5deg grid is fine but you have to say that this will deteriorate the skill of the coarser grid spacing simulations.

L228: Temperature can have strong gradients along coastlines and in orographically complex regions that you mention quite a lot in your results. Therefore, using bilinear interpolation might also not be the best choice here.

L228: I agree that these are rare events but you should mention that you decrease your sample size by looking at rare events, which makes robust statistics more challenging.

L251: I am not familiar with this method of pooling extreme values from an ensemble. You correct the models for a mean bias but you do not correct for the shape and scale

of the tail of the distribution. Does this not mean that you base your high return values mainly on the models that have a very long tale? Is there a reference for this method?

L283-7 & 304-8 & 318-9 & 364-8: This information does not have to be duplicated here since it is already mentioned in the figure caption.

L332-3: Gauge under catch could not only contribute it definitively does. This can be substantial especially for extreme precipitation that are associated with high wind speed (northern latitudes) and in snow-dominated regimes.

L474: I can also see this in the N96 simulation but the Alps are much wider at this grid spacing.

L481-3: A model's grid spacing is always higher than its resolution (see e.g. Skamarock 2014). L559: Did you detrend your 500 hPa geopotential height before you did the analog analysis. There is a high chance that the 500 hPa geopotential increased during your simulation period, which might affect your analog analysis.

L542-73: This should go into the methods section.

L696-7: Getting benefits from upscaling might demand convection-permitting climate simulations (Hart et al. 2018).

L715: This is not true in Ban et al. (2015). They show very similar increases in extreme precipitation between their 12 km and 2 km model results.

Literature:

Skamarock, W.C., 2004. Evaluating mesoscale NWP models using kinetic energy spectra. Monthly weather review, 132(12), pp.3019-3032.

Hart, N.C., Washington, R. and Stratton, R.A., 2018. Stronger local overturning in convective‐permitting regional climate model improves simulation of the subtropical annual cycle. Geophysical Research Letters, 45(20), pp.11-334.

---

## Author Comment (AC1) · 17 Mar 2020

The paper, in general, is well written and the authors attempt to solve a very critical issue regarding the added value of increasing resolution. However I do have some issues regarding the dataset used and some of the methodologies applied, hence I recommend major revision.

Thank you for your comments and suggestions. Please find our point by point responses below.

[Figure]

Here my major comments:

a) All the datasets used to compare the models' output are labelled as observations when, in reality, they are not, they are reanalyses. So I wonder why the authors chose a reanalysis when there are real observational datasets, like GPCP, CHIRP or TRMM. I do know all those datasets have some caveats, but they are derived from direct observations, (and most of them are of the same resolution like the one that the reanalysis used in the paper). At least for precipitation and temperature, I suggest that the authors re-do the comparisons with some of the observational datasets mentioned. For wind, I understand that there might not be another option that the one that they used.

In fact, for temperature and precipitation we primarily use E-OBS, which is a gridded station based dataset. The MESAN reanalysis for precipitation is not a typical reanalysis, in that it is adjusted using station measurements of precipitation afterwards. We now make the text clearer on this (see below). For wind, yes, it is true that reanalyses were the only readily available option. However, we have now changed the reanalyses used for wind in response to reviewer 2's comments to DYNAD (which is related to MESAN), MESCAN and ERA5. The first two are 5.5km resolution downscalings of the 22km resolution HIRLAM and 11 km UERRA-HARMONIE reanalyses respectively (see section 2.1 for details). ERA5 was not available when we first conducted the analysis, but we now use it to replace the ERA-interim based wind estimates (ECEM and WFDEI). We have now been more careful in the text to specify where we mean observations and where we mean reanalyses.

"results are repeated for precipitation extremes using the 5.5 km resolution MESAN reanalysis (Landelius et al. 2016), which adjusts a downscaled first guess from the 22km resolution HIRLAM reanalysis (Dahlgren et al. 2016) with a network of station-based precipitation observations."

b) I do not understand why the authors chose to use climatological means of Txx and Rx1day. Would not be more useful to have seasonal maps? Mostly for precipitation,

as extremes, can occur either in summer or in winter but by very different processes. In this way, we could have seen which type of seasonal extremes are better (or worse) capture and this might also complement the results of the analogues.

Due to space constraints we are not able to show seasons. For wind and temperature this is unlikely to matter (since heatwaves are a summer phenomenon), but for precipitation it might make a difference. We attempted to examine precipitation extremes arising from different processes by examining both Rx1day and Rx5day, the former being more likely to represent convective thunderstorms, and the latter more large scale precipitation (although it is not a perfect division, but seasons would not be either). We did not find large differences between the results for Rx1day and Rx5day, so we only show the former, and mention the latter in some places.

c) On the same idea, why to use a 5-day average (line 568) mainly for precipitation, given the fact that by doing that you are smoothing the extreme event that normally would last one day. For temperature, it might not be a problem, considering that heatwaves per definition last several days(at least 3).

In fact, in the paper we show 1 day precipitation (Rx1day) and 5 day temperature. 5 day precipitation was also examined, but is not shown, although the results are mentioned in a couple of places in the text. We now make this clearer by adding this sentence at line 242 "Rx1day and TXx5day are presented in the figures, whilst the other indices are commented on in the text." ("other indices" refers to TXx and Rx5day).

General comments:

1. Line 40, please define what do you mean by "small scale", do you mean synoptic, mesoscale?

We have added :" i.e on the order of a few km to a few hundred km"

2. Line 60, remove "really"

Done

3. Line 62, added to the list of reference: Risanto, C.B.; Castro, C.L.; Moker, J.M., Jr.; Arellano, A.F., Jr.; Adams, D.K.; Fierro, L.M.; Minjarez Sosa, C.M. Evaluating Forecast Skills of Moisture from Convective-Permitting WRF-ARWModel during 2017 North American Monsoon Season. Atmosphere 2019, 10, 694.

We have now added this reference.

4. Line 66-67, I think the justification of centre this paper on Europe, has to other than" "its climate is highly variable and affected by a range of both large and small scale processes which present challenges for adequate simulation", as this is true for several regions in the planet.

The reviewer is correct. This statement was more of an aside point, rather than the main motivation. Europe also has a large number of coordinated RCM simulations at two different resolutions as part of the EUROCORDEX project, which lend themselves to this kind of analysis. We edit the existing sentence as follows:

We will address these questions focusing on Europe, for which a large number of coordinated RCM simulations at two standard resolutions are available as part of the EUROCORDEX project (Jacob et al., 2014), and whose climate is highly variable and affected by a range of both large and small scale processes, which present challenges for adequate simulation.

5. Line 56: " : : :history and trajectory of air masses ARE important : : : (instead of"is")

Done

Anonymous Referee #2

In the manuscript "The benefits of increasing resolution in global and regional climate simulations for European climate extremes" Iles et al. assess the dependence of simulating extreme event intensity on model resolution and model strategy (regional vs. global) over Europe. They show that higher resolution simulations have generally stronger extremes with higher sensitivities for precipitation and wind extremes than

for temperature extremes. Their results generally confirm previous studies and for me, the greatest value of this paper is the combination of a large range of models and investigating three extreme events, which provides a good overview. I also like the upscaling analysis although it is kept fairly brief. However, I have concerns in how the team compared models with different resolutions and would like to see a more careful usage of the word model bias especially in the context of precipitation and wind analysis for which the used observational datasets are of questionable quality. Below is a more detailed description of my major/general and minor comments.

We would like to thank the reviewer for his/her very thorough and constructive review. Please find our point by point responses below.

Major/general comments:

1. I have concerns with using nearest neighbor remapping for extreme value analysis. This remapping method will artificially increase extremes in high-resolution data particularly in situations with strong gradients. For instance, precipitation extremes are typically very localized and have strong spatial gradients. You are remapping the 0.11 EURO-CORDEX simulations to a 0.5deg grid. This means that you pick the closest 0.11deg grid cell to each 0.5deg cell and assume that it rains ass much on the 0.5deg area than on the 0.11deg area. This is certainly not the case and by doing this, you violate mass conservation of precipitation. I strongly recommend using a conservative remapping method that conserves mass and energy while remapping. This will affect many of your results since you generally find that extremes are more intense in the high-resolution simulations. Using conservative remapping will dampen this effect and might change your conclusions.

We agree with the reviewer that using nearest neighbour is not the most appropriate technique for going from high to low resolution, as we only use information from a small subset of grid cells. This can indeed be problematic close to strong gradients. But conservative remapping made the extremes much weaker than any other technique,

especially for CORDEX 0.11 (see figure S1) and there seems to be a dampening effect not only from averaging over larger areas (as we expect), but also a further dampening from splitting grid cells that fall on the boundaries of the new grid into two or more- which happens to a large proportion of grid cells. We also see this dampening effect of conservative remapping when going from low to high resolution. We therefore changed the regridding for cordex 0.11 (and similar resolution observational datasets) to bicubic interpolation, since this also replicated the results using the original grid well, whilst using information from all grid cells.

So in summary, we decided to use bicubic interpolation for going from high to low resolution (for CORDEX 0.11, MESAN and MESCAN and ERA5) and nearest neighbour for everything else. Figure S1 shows that this decision (to use nearest neighbor) seems appropriate even for medium resolutions (e.g. CORDEX 0.44 and UPSCALE 25km (N512)).

2. Along coastlines, you have to be careful that you only consider land grid points in your evaluation against land-based observations. As you mention in your wind speed analysis, there is a sharp gradient in wind speed but also for temperature extremes between ocean and land. I am not sure if your interpretation that high wind extremes propagate further inland in coarse-scale models is correct. Could it be that you include ocean grid cells in your analysis, which cause a high bias in low-resolution models?

On closer inspection, we agree that the mask used for wind included too many ocean grid points. This was based on the mask of the WFDEI dataset. Instead we have applied the E-Obs mask (which was also used for temperature and precipitation), which you can see in figure 4 approximates the land much better. This does affect some of the results for wind, and we thank the reviewer for pointing this out. Since model land masks contain information on land area fractions rather than binary land/not land we expect that the coastal grid cells we refer to are a mixture of land and ocean – this still has the effect of making ocean influences appear further inland – just by nature of the grid box overlapping land and ocean areas more- this is what we meant by "propagate

further inland". We have made this clearer in the text and added the possibility of there being other differences in the land masks with resolution. We prefer not to select only grid cells that are 100% land from each model (leading to a different land area for each model) since the increasing detail and accuracy of a model's land mask is a fundamental advantage of increased resolution and correcting for it constitutes a sort of bias adjustment. Inaccurate model land masks also have implications for local scale climate impact management decisions.

3. Please use a lapse rate adjustment when comparing temperature extremes between model grid spacings. Your finding that heat extremes are less overestimated in high-resolution simulations in topographic regions is trivial since the coarse resolution models have lower topography and therefore higher temperatures. You could simply use the topography of E-OBS and correct the model temperature with a climatological average lapse rate (e.g., 6.5 deg C per km).

We think that the higher topography associated with higher resolution is a fundamental benefit of increasing resolution, just as the lower topography in low resolution models is a fundamental drawback. Correcting for differences in topography is a kind of bias correction. We prefer to present the models as they are, without such adjustments.

4. Please avoid writing about model biases when you compare extreme precipitation to E-OBS. You mention that E-OBS underestimates precipitation extremes, which means that you would like a model to be wetter than E-OBS. Evan the MESAN precipitation extremes are likely too low. I suggest being very careful how you use the word bias and rather use difference when comparing to precipitation but also wind datasets.

Good point. We have now changed the language in these sections to reflect that these are not necessarily biases, but differences.

5. The separation of CMIP5 and CORDEX results from the UPSCALE results make the paper unnecessary long and harder to read. I recommend combining sections 4.1 and 4.2 since I do not see a good reason to separate the UPSCALE ensemble from the

other datasets. Also, combining figures 1, 3, and 4 with 5, 7, and 8 would be beneficial. You could do this by not showing the results for all and common ensemble members but only one of the two. They are very similar anyway.

We prefer to keep these two sections separate. They are in fact addressing different questions. The CORDEX vs CMIP5 analysis is looking at the benefits of regional dynamical downscaling, whilst the UPSCALE analysis addresses the effects of increasing resolution globally. We feel that combining the sections would make the paper less clear to follow.

6. Also, the summary and discussion section could be combined. After the summary section you again briefly summarize results in the discussion. You would lose very little information by removing the summary section.

We have now combined these two sections.

7. I am concerned with how different the three wind observations are. Are they all equally likely? Looking at the big differences between these observations makes me wonder if you can/should evaluate the models at all. Just looking at your fields in Fig. 4 (c,f,i) makes me wonder if these datasets can even capture topographic effects. The low wind extreme minimum over the Alps looks very unrealistic and event he models seem to do a better job capturing these effects that the observations. Would the use of MESAN winds be a better option?

We agree with the reviewer that the ECEM dataset with the Weibull distribution based bias correction seems unrealistic (this bias correction technique may have worked for mean winds, but not for the extremes). We have opted to replace all three datasets. We now use MESAN winds (actually called DYNAD), as suggested by the reviewer, but also MESCAN (which like MESAN is available at 5.5 km resolution, and is constructed by downscaling of the 11km UERRA-HARMONIE reanalysis with a NWP) and ERA5. The spread amongst the datasets is now much less, although still present. ERA5 seems to have particularly slow winds, including over the Alps, but we show it since it

is a new reanalysis that many people are likely to make use of.

8. Please be careful when you use the term model resolution and do not confuse it with model horizontal grid spacing. The model resolution is typically 4 to 8 times the model horizontal grid spacing (Skamarock 2014)

We have now been more careful in the introduction to clarify when we actually mean horizontal grid spacing rather than resolution, and have also added a clarifying sentence at line 72 stating "Throughout the rest of this manuscript we use the term "resolution" to mean model horizontal grid spacing, whilst recognising that a model's effective resolution, in terms of the scales it can capture, is always less than its grid spacing."

L37-8: Please explain what a long-standing anticyclone is? Do you mean a stationary anticyclone?

Yes, we have renamed it "stationary anticyclone"

L41: I would say "flash floods" here since river floods involve large-scale processes.

Agreed, we have changed the text accordingly.

L43: I suggest "poorly resolved" since some of these processes can be resolved at fairly large grid spacings.

Agreed, we have changed the sentence to "These are poorly resolved at the resolution of Global Climate Models (GCMs) in CMIP5 (Coupled Model Intercomparison Project Phase 5; Taylor et al., 2012)"

L50: Could you please be more specific than "small-scale processes and features involved"

We have added the following (in bold): "For precipitation and wind extremes, an improvement with resolution could be expected due to the small-scale processes and features involved, including convection and the influence of topography."

L89: What do you mean with "their" in "to their information"?

"Their" refered to "heatwaves". We have now changed the text to explicitly say "heat-waves"

L136: E-OBS has quite high station density in some regions (e.g., Slovenia, Germany) but low density in others (e.g., Austria, Spain)

This is a good point. We have updated the sentence to the following: E-OBS has a somewhat non-uniform underlying station density, with relatively high densities in Germany, Sweden and Slovenia, and low densities in other countries (e.g. Spain, France, Austria). It tends to underestimate precipitation extremes relative to higher density regional datasets, especially where it has poor coverage, due to missed extremes which are local in scale (Prein and Gobiet 2017)."

L168L: "us to identify"

Corrected

L173: Are these daily maximum surface wind speeds based on model time step wind speed or hourly maxima of instantaneous wind, or something else?

The variable sfcWindmax in model outputs is based on the model time step wind speed, although figure S7 (which compares return periods of annual maximum wind based on sfcWIndmax with that based on 3 hourly and 6 hourly data for the GCMs that have both) suggests that this is not the case for the IPSL models or CMCC-CM for which annual maximum sfcWindmax has slower speeds than both 3 and 6 hourly estimates. This difference in definition between models is a weakness of the analysis. Also, since we expect the model time step to decrease with increased resolution, we would expect this to result in stronger winds with higher resolution due to the increased sampling frequency. Whilst in some ways this latter point makes the models not strictly comparable, being able to generate stronger winds due to a shorter time step would nevertheless be an inherent feature of higher resolution models.

[Figure]

Comparing all models using 3 hourly (or 6 hourly) data would have been tidier, but this data was simply not available for CORDEX 0.44 and very limited for CORDEX 0.11. These caveats and their implications are made clear in the text.

To the former line 173 we have added "This seems to mostly be based on model timestep wind speed, with a few exceptions (see figure S7). The implications of this are discussed further in the results section." And to the results section we have added some discussion about differences in model time steps.

L174: "Those consist of"

We changed to "These consist of"

L192: What is this alternative dataset for SST?

It was the OSTIA analysis (Operational Sea Surface Temperature and Sea Ice Analysis) (Donlon et al. 2012). We have added this to the text.

L203: Performing your analysis on a 0.5deg grid is fine but you have to say that this will deteriorate the skill of the coarser grid spacing simulations.

We have added further discussion of these points. This paragraph now reads (updates in bold)

"In order to compare models of different resolutions with each other and with observations it was necessary to regrid variables to a common grid. Using a high resolution grid for evaluation would preserve the finer spatial detail and localised extremes for high resolution simulations, but is sometimes considered unfair for coarse resolution models which cannot be expected to simulate the same intensities of extremes even for a perfect simulation due to spatial smoothing effects (Prien et al. 2016). However, the finer spatial detail is an inherent advantage of high resolution and smoothing this out will result in information loss. We use a 0.5° regular longitude-latitude grid since it is in-between the resolution of the CORDEX models and CMIP5, is computationally feasible and E-OBS is also available at this resolution. Some of the benefits of higher

resolution may be lost by doing this, putting our results on the conservative side. Nevertheless, sensitivity tests showed that results for MESAN did not change perceptibly by using a 0.5° grid compared to a 0.1° regular grid. We regrid the daily data, before the calculation of annual extreme indices. "

L228: Temperature can have strong gradients along coastlines and in orographically complex regions that you mention quite a lot in your results. Therefore, using bilinear interpolation might also not be the best choice here.

Thank you for raising this point. We agree that using bilinear interpolation will falsely smooth some features. Having now also repeated the sensitivity analysis to regridding that we did for precipitation (fig S1) for temperature, we found a similar (but reduced) sensitivity of temperature extremes to regridding method. Bilinear interpolation reduced the return values seen in the return period plots. We found that the choices we made for precipitation and wind (nearest neighbour for regridding from low to high resolution, or between similar resolutions, and bicubic for high to low resolution) were also the best choices for temperature in terms of replicating the return curves that we get by using the original grid, whilst also preserving the blocky nature of the spatial patterns from the lower resolution models.

L228: I agree that these are rare events but you should mention that you decrease your sample size by looking at rare events, which makes robust statistics more challenging.

We add "One drawback is that this makes robust statistics more challenging."

L251: I am not familiar with this method of pooling extreme values from an ensemble. You correct the models for a mean bias but you do not correct for the shape and scale of the tail of the distribution. Does this not mean that you base your high return values mainly on the models that have a very long tale? Is there a reference for this method?

We have now replaced the results based on pooling with ensemble means. This was because we realized that pooling interacted with the spatial averaging to change the

Interactive
comment

shape of the distributions (especially for precipitation), making them no longer compa-
rable to the observations. This was also the case for the UPSCALE simulations where
there was no issue with different models being combined in the pooling. Instead we
use the ensemble mean or median, which retains comparability to the observations.

Regarding the distribution of the models in the pooled distributions, figure S3 (in the
original submitted version, now removed) showed that the tails were not dominated
by any one model, although models were not spread 100% evenly across the pooled
distribution either

L283-7 & 304-8 & 318-9 & 364-8: This information does not have to be duplicated here
since it is already mentioned in the figure caption.

This information has now been deleted or shortened in these places.

L332-3: Gauge under catch could not only contribute it definitively does. This can
be substantial especially for extreme precipitation that are associated with high wind
speed (northern latitudes) and in snow-dominated regimes.

Added: "Gauge undercatch will also contribute to the difference, particularly for precip-
itation extremes associated with strong winds and in snow dominated regions"

L474: I can also see this in the N96 simulation but the Alps are much wider at this grid
spacing.

We change the wording to: "All resolutions have bands of heavy precipitation either
side of the Alps, but these move closer together as the Alps become better defined"

L481-3: A model's grid spacing is always higher than its resolution (see e.g. Skamarock
2014).

Thank you. We have now added a sentence on this at line 72 (see response to com-
ment 8 above). We have also updated this sentence to "In addition, it should be noted
that models with the same nominal resolution do not necessarily have the same effective resolution, and that the effective resolution is always less than the nominal resolution." The main point we wanted to communicate here was that just because two models claim to be the same or similar resolutions in terms of grid spacing, it doesn't mean they are the same in terms of the actual scales that can be resolved.

L559: Did you detrend your 500 hPa geopotential height before you did the analog analysis. There is a high chance that the 500 hPa geopotential increased during your simulation period, which might affect your analog analysis.

500 hPa was not detrended in the original analysis. We have re-run the analogue analysis and replaced the relevant figures using pattern correlations as the measure of distance between circulation states instead of Euclidean distance. This should get around issues relating to trends in geopotential height. Results were not sensitive to this change in method. We have updated the text accordingly: "Similarity between circulation states is quantified using pattern correlation, which is not affected by trends in geopotential height with global warming"

L542-73: This should go into the methods section.

We think that it improves the readability of the paper to keep the description of the circulation analogues method here. The circulation analogue analysis builds on the analysis presented in the rest of the paper, and we feel that moving the description to the methods section would overload the reader, forcing them to imagine many steps that become more concrete after seeing the figures in the rest of the paper.

L696-7: Getting benefits from upscaling might demand convection-permitting climate simulations (Hart et al. 2018).

Thank you, we have added this citation.

L715: This is not true in Ban et al. (2015). They show very similar increases in extreme precipitation between their 12 km and 2 km model results.

This is true for daily precipitation, but they do see a greater increase for sub-daily

summer precipitation in the 2km model. We have modified this sentence to specify only summer sub-daily intensities (rather than both daily and sub-daily), and made the sentence read less definitively.

Literature:

Skamarock, W.C., 2004. Evaluating mesoscale NWP models using kinetic energy spectra. Monthly weather review, 132(12), pp.3019-3032.

Hart, N.C., Washington, R. and Stratton, R.A., 2018. Stronger local overturning in convectiveâËŸARËĞ permitting regional climate model improves simulation of the subtropical annual cycle. Geophysical Research Letters, 45(20), pp.11-334.
* * *

---

## Referee Report (RR1)

This paper examines benefits of increasing resolutions in CMIP5 GCMs and EURO-CORDEX RCMs on extreme temperature, precipitation and wind indices. The paper examines a critical topic in the field of model development. In my opinion the paper serves to what was proposed. However, I have a couple of major comments about the methodology and a few of minor comments about the manuscript. I would strongly encourage the authors to add at least a qualitative discussion on the points mentioned below.

Major comments:

1) Authors have used empirically computed return periods. This method of computing return periods has a major limitation that it only considers the rank and not the actual magnitude of the data. Therefore, the largest return period computed here from 36 years of data cannot exceed 36 years. The method does not calculate return periods with sufficient accuracy in some cases such as a trend in the data or passage of a single storm of unusually very high intensity. In such cases the return periods will be underestimated. A proper way of estimating return periods is to fit a theoretical generalized extreme value (GEV) distribution to block maxima (annual maxima, here), and then compute return periods from using the parameters of the fitted distribution.
I would encourage authors to recognize this aspect of the limitation in their methodology and associated impact on return periods.

2) Authors have bias adjusted the data before computing return periods. This artificially reduces/ enhances the model maxima to appear closer to the observational estimates. In my opinion this hides the "true" model performance. For example, a comparison of Fig. 6 and S9 suggests that models perform poorer when evaluated without bias-correction. In my opinion an objective model estimation should not include bias-correction. Bias-correction should only be used after a model has been evaluated.

3) Authors have used "multimodel means". Multimodel means could strongly be affected by one or two outlier model. Instead, multimodel medians are more robust in a sense that they are rather insensitive to any outliers. I would encourage authors to discuss this limitation in their manuscript.

4) Also, it appears that authors have used all ensemble members from a model to compute "MM all" as in figures such as Fig. 1, 3 etc. This method of computing multimodel means gives more weight to a model with multiple ensemble members than a model with a single or a smaller number of ensemble members. This will also most likely result into model biases that are not representative of "common" model biases across different sets of GCMs.

Minor comments:

1) At several places (e.g., lines L201, 251, 265, 281 etc.) authors have used the term 'observation/s" for observational datasets (E-OBS, MESCAN etc.). These observational datasets are not "observations".
2) L206: What is OSTIA?
3) L80: "Precipitation extremes tend to get heavier and agree better with observations". When? With increasing resolution? If yes, please mention this.
4) L77-79: I do not understand this sentence completely. What do you mean by "for grid point models"? I suggest authors to use short and simple sentences instead of a long complex sentence (e.g., L477-480).
5) L318-319: The differences will be bigger when return periods are computed without bias-adjustment.
6) L323:324: This statement does not seem to be completely true. Cold biases in Scandinavian region increase considerably from CMIP5 to CORDEX 0.44. Also, the warm bias in the eastern part of Central Europe has decreased from CMIP5 to CORDEX. The "insensitivity" observed in Fig. 2 may be partly due to bias-adjustment.
7) Fig. 5 has color scales swapped between mean and bias figures. As a general comment, I would highly encourage authors to use different color schemes for representing totals and biases. Using once color scheme for both is very confusing while examining many figures.
8) L470-471: Return level plots are not distribution plots. Shape of the annual maximum can only be examined by estimating shape parameters of the fitted GEV distribution. Please correct this sentence.
9) L589-590: Are you referring to Fig. 6 here? If yes, please mention this for clarity.
10) L590-592: It appears that for Rx1day downscaling is more dominant over 3 out of 5 regions examined.
11) Table S1: Replace "donate" with "denote".
12) Table S1: "with those forming part of the "common subset" in **bold**. I think "bold" should be replaced by "colors".

---

## Author Response (AR2)

**Reviewer1**

**MS No.: gmd-2019-253**

This paper examines benefits of increasing resolutions in CMIP5 GCMs and EURO-CORDEX RCMs on extreme temperature, precipitation and wind indices. The paper examines a critical topic in the field of model development. In my opinion the paper serves to what was proposed. However, I have a couple of major comments about the methodology and a few of minor comments about the manuscript. I would strongly encourage the authors to add at least a qualitative discussion on the points mentioned below.

*Thank you for your helpful comments and suggestions. Please find our point by point responses below.*

Major comments:

1) Authors have used empirically computed return periods. This method of computing return periods has a major limitation that it only considers the rank and not the actual magnitude of the data. Therefore, the largest return period computed here from 36 years of data cannot exceed 36 years. The method does not calculate return periods with sufficient accuracy in some cases such as a trend in the data or passage of a single storm of unusually very high intensity. In such cases the return periods will be underestimated. A proper way of estimating return periods is to fit a theoretical generalized extreme value (GEV) distribution to block maxima (annual maxima, here), and then compute return periods from using the parameters of the fitted distribution.

I would encourage authors to recognize this aspect of the limitation in their methodology and associated impact on return periods.

*We add this sentence to the methods section. "This is an empirical approach and has the limitation that return periods cannot exceed the number of years of data used (e.g. 36 years). This is still the case even if an extremely unusual event occurs. Using a GEV would allow estimates for higher return periods, but this would still be an extrapolation."*

*Also, a GEV would also be affected by non-stationarity in climate.*

2) Authors have bias adjusted the data before computing return periods. This artificially reduces/ enhances the model maxima to appear closer to the observational estimates. In my opinion this hides the "true" model performance. For example, a comparison of Fig. 6 and S9 suggests that models perform poorer when evaluated without bias-correction. In my opinion an objective model estimation should not include bias-correction. Bias correction should only be used after a model has been evaluated.

*Biases of the indices used are shown in the map plots. The focus of the return period plots is instead on the shape of the return curves. For CMIP5 and CORDEX, the large variety of models used gives rise to a large spread of curves due to mean biases in the indices which makes it very difficult to compare the shape of the curves between models, and also with the observations. The adjustment that we apply aims merely to shift the curves up or down so that they have the same mean in order to allow differences in their shape to be seen. We do not aim to perform a formal bias adjustment. This was not such an issue for the UPSCALE simulations, being based on a single model version per resolution which causes curves to be tightly clustered anyway. (This is why we show results without such an adjustment in the main text, and with the adjustment in the supplement). We have re-written the relevant part of the methods in response to reviewer 2 to make all these points clearer.*

3) Authors have used "multimodel means". Multimodel means could strongly be affected by one or two outlier model. Instead, multimodel medians are more robust in a sense that they are rather insensitive to any outliers. I would encourage authors to discuss this limitation in their manuscript.

*The multi model means have now been replaced by multi-model medians for CORDEX and CMIP5. For UPSCALE the number of simulations was small (3 or 5 per resolution) and all came from the same model (meaning outliers are unlikely), so we kept the means.*

4) Also, it appears that authors have used all ensemble members from a model to compute "MM all" as in figures such as Fig. 1, 3 etc. This method of computing multimodel means gives more weight to a model with multiple ensemble members than a model with a single or a smaller number of ensemble members. This will also most likely result into model biases that are not representative of "common" model biases across different sets of GCMs.

*This is a valid point. We have now replaced "MM all" results using one member per model both for CMIP5 and CORDEX.*

Minor comments:

1) At several places (e.g., lines L201, 251, 265, 281 etc.) authors have used the term 'observation/s" for observational datasets (E-OBS, MESCAN etc.). These observational datasets are not "observations".

*"Observations" has now been replaced by "observational dataset" throughout.*

2) L206: What is OSTIA?

*Operational Sea Surface Temperature and Sea Ice Analysis. The longer name has been added to the text.*

3) L80: "Precipitation extremes tend to get heavier and agree better with observations". When? With increasing resolution? If yes, please mention this.

*Yes, this is now said explicitly*

4) L77-79: I do not understand this sentence completely. What do you mean by "for grid point models"? I suggest authors to use short and simple sentences instead of a long complex sentence (e.g., L477-480).

*"Grid point model" is a standard term describing GCMs that perform calculations on a grid and is in contrast to "spectral models". The sentence has been adjusted as follows: "In GCMs, global precipitation tends to increase with resolution, and for grid point GCMs (as opposed to spectral GCMs) the fraction of land precipitation and moisture fluxes from land to ocean increases, largely due to better resolved orography.*

*The complex sentence you refer to has been split into two.*

5) L318-319: The differences will be bigger when return periods are computed without bias adjustment.

*We are really focusing on the difference shapes of the tails in the return period plots rather than differences in the mean values of the indices (which are shown in the map figures). The purpose of the "bias correction" is not really to perform bias correction in itself, we merely wish to shift all the curves together to have the same mean in order to be able to compare their shapes, which would not be possible when their mean values are totally different. The description of the bias correction has been reworded to reflect this.*

6) L323:324: This statement does not seem to be completely true. Cold biases in Scandinavian region increase considerably from CMIP5 to CORDEX 0.44. Also, the warm bias in the eastern part of Central Europe has decreased from CMIP5 to CORDEX. The "insensitivity" observed in Fig. 2 may be partly due to bias-adjustment.

*You are correct concerning the Scandinavian cold bias. The difference in the warm bias between CMIP5 and CORDEX in the "common subset" of simulations has decreased now that we use ensemble medians instead of means. This sentence has been rephrased as follows:*

*In summary, shapes of return period curves for temperature extremes appear to be insensitive to dynamical downscaling based on comparing CMIP5 to CORDEX at 0.11° and 0.44°, but biases are affected, for instance over mountains where hot biases decrease with resolution.*

*I do not specifically mention this cold bias in this summary statement, because it is unclear whether it is resolution related, or due to other causes, and it does not get worse in the 0.11° simulations compared to the 0.44° ones. However, it is now mention in the discussion section. See also the answer about biases above.*

7) Fig. 5 has color scales swapped between mean and bias figures. As a general comment, I would highly encourage authors to use different color schemes for representing totals and biases. Using once color scheme for both is very confusing while examining many figures.

*Thank you for pointing out that the colour bars were the wrong way round. This has now been corrected. Whilst we appreciate the comment that using a different colour scheme for the biases might be less confusing, this would be very time consuming to implement due to the large number of figures (especially in the supplement- the figures with many panels are extremely slow to plot and adjust).*

8) L470-471: Return level plots are not distribution plots. Shape of the annual maximum can only be examined by estimating shape parameters of the fitted GEV distribution. Please correct this sentence.

*Apologies, we were referring to the shape of the return period curves rather than a formal shape parameter. Nevertheless, it is still possible to comment on whether or not the tails of the distribution get heavier with resolution or not based on these curves. We have corrected this sentence : "*whilst the shapes of the return period curves are insensitive"

9) L589-590: Are you referring to Fig. 6 here? If yes, please mention this for clarity.

*No, all this can be seen in Figure 9.*

10) L590-592: It appears that for Rx1day downscaling is more dominant over 3 out of 5 regions examined.

*These sentences have been rephrased.*

11) Table S1: Replace “donate” with “denote”.

*Thank you for pointing this out.*

12) Table S1: “with those forming part of the “common subset” in **bold**. I think “bold” should be replaced by “colors”.

*Agreed*

**Comments to: gmd-2019-253, Iles et al. “The benefits of increasing resolution in global and regional climate simulations for European climate extremes”.**

Overall recommendation: minor revision.

This is an interesting paper about the effect of increasing model resolution on extreme events, considering the added value of regional climate models with respect to the driving GCMs and different spatial resolutions for the same GCM. An analysis assigning the differences due to resolution to upscaling or downscaling effects is certainly interesting, although a bit too succinct. The first part of the paper makes use of an impressive set of simulations and the consideration of observational uncertainty in the evaluation of precipitation and wind speed is highly appreciated. The paper reads well and is well-structured. It adds valuable insights to the existing literature, which is nicely referenced in the discussion.

I would recommend the consideration for publication in Geoscientific Model Development after minor revision. Note that although labeled as minor, these issues are relevant and should be well addressed in a revised version of the manuscript.

*Thank you for your thorough review and your helpful comments and suggestions. Please find our point by point responses below.*

My main concerns are:

1) Bias correction. Why was the bias correction applied directly to the indices instead to the daily input data before obtaining the indices (L278-279)? I think that a correction based on the indices themselves could be noisier since they are values in the upper tail of the distributions and the objective is to look at return values which might be even more sensitive to unstable corrections. Applying a simple correction, such as the mean of the daily distribution, prior to the indices calculation would be more robust and also better for consistency among the indices based on the same variable. As far as I understand, maps show biases of the raw data (that should be said explicitly) but bias-corrected values values were only used for the return values (that should be stressed), thus as expected, moving the models towards the observations. It should be clearly motivated why this is the case or done in a more appropriate way. I also do not understand why bias correction was not applied to the UPSCALE simulations because of being only one model (L455-457). I think these simulations should be corrected as well, due to the nature of the analyzed metrics and for the sake of comparability with the previous sections.

Also, was it also an additive correction for precipitation and wind indices? I would expect to have a multiplicative correction in such cases.

*Using the word “bias adjustment” as we did was misleading. In actual fact we did not really apply a formal bias adjustment as such- the only thing we wished to do was to shift the return period curves up or down to have the same mean value of the index in question so that we could focus on differences in the shapes curves (since biases are already shown in the map figures). Without being shifted in this way, the curves for different models are so spread out that differences in the shapes of the curves (for CORDEX and*

*CMIP5) are impossible to examine. This was not so critical for the UPSCALE simulations because the same model is used for a given resolution, and the curves tend to naturally cluster tightly together, allowing both differences in shape and mean biases to be seen adequately in the figures. However, we also show the adjusted versions in the supplement for comparison.*

*We update the text accordingly: "In order to allow the shapes of the return period curves to be compared more easily between different types of models (i.e CMIP5 and CORDEX at both resolutions), we first adjust each model to have the same climatological mean value of the extreme index in question. This effectively shifts the curves up or down, but does not change their shape, which is the focus of these figures. Without such a shift, curves are too spread out to be able to discern differences in shape. Therefore we cannot comment on mean biases of the extremes indices based on the return plots, but these biases are already shown and discussed based on map figures (see section 3.1). We implement this adjustment by subtracting the difference between the model climatology of the index in question and the climatology of the reference observational dataset for each model at a grid cell level. We use E-OBS as the reference for temperature and precipitation, and MESCAN for wind. The additional observational datasets shown in the return period plots are also adjusted in the same way. For the UPSCALE simulations, results can also be examined without the need to shift the curves to a common mean value because the same version of the same model is used for a given resolution, meaning that curves for individual simulations tend to cluster together instead of having large mean differences. In this way, differences in biases with resolution are also seen in the return period plots. Nevertheless, we also present UPSCALE results with the adjustment in Figure S9 for comparison."*

2) Inconsistencies in wind extremes (L374-376). I acknowledge the explanations and the sensitivity analysis carried out about the different temporal resolution of the wind speed variables in the models. However, I would recommend not to include the analysis with such caveats. A safer way to go would be to consider for wind extremes only the models which can provide the variable which is consistent with the observations (6-hourly). I think that this consistency is more important than keeping consistency with the temperature and precipitation extremes, or than having a larger ensemble. Also, I do not understand the reasoning that values depend on the timestep; wouldn't the primary time step for a given model the same for the three variables? The differences for CMCC, CNRM and the IPSLs are massive in the sensitivity analysis.

*Since the last version of this paper a lot more 3 hourly wind data became available on ESGF for CORDEX at 0.11°. Therefore, we have replaced the analysis that was based on sfcWindmax with an analysis based on 3 hourly wind comparing CMIP5 to CORDEX 0.11° (The three hourly data are subsampled to 6 hourly by taking every second value). This is now consistent with the reanalysis datasets and avoids inconsistencies in the way sfcWindmax is computed between models. CORDEX at 0.44° could not be included for the main analysis because there was no overlap between the GCM-RCM combinations used for the two different CORDEX resolutions (and all five 0.44° simulations used RCA as the regional model). In order to allow a comparison of CORDEX at 0.11° and 0.44°, Figure S8 and S9 show a comparison of sfcWindmax for models with data at both resolutions (9 models). sfcWindmax should be computed in a consistent way between the same CORDEX simulations at different resolutions and so is free from some*

*of the caveats previously mentioned. All text relating to the wind analysis in CORDEX and CMIP5 has been updated.*

*Concerning the influence of the timestep on sfcWindmax- wind is an instantaneous value recorded at each model time step. SfcWindmax is the maximum of these values per day. The smaller the timestep, the less chance of peak wind speeds being missed due to the sampling frequency. This affect is illustrated by the comparison of 3 hourly and 6 hourly winds in the former figure S7 in the previous version of this manuscript. Annual maximum wind based on 3 hourly data gives stronger wind speeds than using 6 hourly data.*

3) Wind reanalyses. It is good to include three reanalyses as reference to sample observational uncertainty (L371). However the differences among the reanalyses for the considered wind extreme index are massive. Are there any studies comparing them, showing how similar are they to real measurements? Which one should we trust more? Their quality should be brought into question: CMIP5 (also true for the UPSCALE simulations) has small bias with respect to ERA5, but that seems to be very unrealistic.

*There are no studies systematically comparing these three reanalysis. However, Jourdier (2020) compared ERA5 to station data for a number of locations in France and found that ERA5 underestimates mean winds, particularly over the mountains. Niermann et al. (2017) evaluated MESCAN compared to German stations and found that extreme wind speeds were too low, whilst slow wind speeds were too fast. Comparing MESCAN and ERA5 in Figure 4 would therefore suggest that ERA5 has an even larger slow bias for extreme winds, whilst DYNAD is similar to MESCAN over Germany. Tomas Landelius who was involved in the creation of both MESCAN and MESAN suggests that the former should be the most accurate. These points have now been added to the manuscript.*

Minor comments

L86-69 When referring to the added value of higher resolution RCMs with respect coarser counterparts, the authors could consider to mention that the added value of the high resolution is not so evident when evaluated on the coarse grid, in particular, the improvement in the spatial pattern of precipitation indices is not statistically significant after applying simple bias correction methods (Casanueva et al. 2016, https://link.springer.com/article/10.1007/s00382-015-2865-x).

*This issue is discussed under the "regridding" subsection. The results found by Casanueva et al. 2016 are also mentioned in the discussion section.*

L102-113 About wind extremes, the authors might consider to mention the added value of coupled regional climate simulations in terms of surface wind and coastal low-level jet (Soares et al. 2019, https://link.springer.com/article/10.1007/s00382-018-4565-9), although this work focuses on northern Africa.

*Thank you for the reference. However, we feel this is a bit too Africa specific.*

L142 Which version of E-OBS was used?

*It was version 15. This has now been specified in the text.*

L 148 A good illustration of the E-OBS limitations (including indices such as return values) can be found in Herrera et al. 2019 (https://essd.copernicus.org/articles/11/1947/2019/) for the Iberian Peninsula.

*Thank you for drawing our attention to this paper. We now cite it here.*

L169-170 I am not sure if "sub-sample" is correct here, since the process goes from 1 hour to 6 hours, wouldn't it rather be "aggregated"? Was the 6-hourly mean or maximum value obtained? Otherwise, please give some further details of the subsampling. Also in L208.

*In this case we take every 6th value in order to go from hourly wind data to 6 hourly wind data. Wind values are instantaneous values. The text has been modified as follows: "We sub-sample ERA5 to 6 hourly data by taking every sixth value in order to be consistent with the other reanalyses."*

L184 It could be worth to mention here that simulations at the two resolutions are carried out with the same model versions and parameterizations, except for REMO, where rain advection is used for 0.11 but not for 0.44 (Kotlarski et al. 2014, https://doi.org/10.5194/gmd-7-1297-2014).

*Thank you, this point has now been added.*

L215 That is probably a too strong statement. Smoothing/upscaling the high resolution might lead to partial information loss, but if there is an added value that might be also present at a coarser resolution.

*We add the word "partial" to this sentence in front of "information loss". We also add another sentence just before: "If processes are captured better at higher resolution, improvements should still be visible when regridded to coarser resolution".*

L270-282 This paragraph does not really fit here. L270-273 was explained already in Sect.2.1.1 (no need to repeat), where the details about EC-EARTHr3 and the combination of the GCMs of the common subset (L274-276) should be moved to. Bias correction (L277-282) does not fit here either, it could be included in a new little subsection after regridding.

*All material about the common subset has now been moved to section 2.1.1 and repetition has been removed. The text about bias correction has been altered as described above and so now fits here.*

L291-293 This paragraph does not fit in the section about return periods. Either this analogue approach is fully described in the Methods in an own section, or this is removed and entirely described in Sect. 4.3. I would go for the second option.

*We have done the second option.*

L313-314 The last sentence of the paragraph is probably the main conclusion of Fig.S3: the driving GCM seems to be the largest source of variability, which is in agreement with previous studies (e.g. Rummukainen, et al. 2001, https://doi.org/10.1007/s003820000109). But I do not understand what the authors mean in this sentence with consistent results for a GCM-RCM chain; consistent with what? The

message is clear if the authors remove "GCM-RCM chain". Also in Fig.3, it would be very helpful to draw a box or mark somehow the columns belonging to the common subset (also in Fig.S5).

*Actually this sentence was referring to different ensemble members of the same model e.g. for CMIP5 this could be EC-Earth r1, r2 and r12. For CORDEX this could be EC-Earth-RCA driven by r1, r3 and r12 of EC-Earth. I have now added a comment on the influence of the driving model on the CORDEX results earlier in the paragraph (new part in bold): "There are also substantial differences between results from different RCMs, including those driven by the same GCM,* ***although the driving GCM does seem to affect the overall magnitude of the temperature extremes****."*

*A box has been drawn around the models that are part of the common subset as suggested.*

Fig.2 is too complicated, I am not able to see the mentioned shadings in the caption, corresponding to the full set or to the subset. Such shadings (if present) could be omitted and I would recommend to show only the individual simulations with different colours and the multi-model median of each subset. Observations and their ranges are a bit difficult to distinguish, the authors could try using another colour.

*By "shade" I was referring to the lightness/darkness of the colour of the multi-model median lines (which is darker for the version based on the common subset and lighter for the version based on all models). I have changed "shades" to "colours" to avoid confusion: "Thin lines are individual ensemble members, thick lines are multi model medians: lighter colours for the full ensembles, and darker colours for the subset of models common to CMIP5 and both CORDEX resolutions." Concerning the observations, I have not managed to think of a colour that shows up better than black.*

L352 What does "models" refer to here? GCMs? RCMs? I think that there is a difference here to the extreme temperature index, since for RX1day results seem to be more consistent for a given RCM regardless of the GCM than for the RCMs with the same driving GCM (see RCA-011).

*Both-but here we refer specifically to either ensemble members of the same GCM (e.g. EC-Earth r1 compared to EC-Earth r12), or the same GCM-RCM chain driven with different members of the same GCM (e.g. EC-Earth-RCA driven by r1, r3 and r12 of EC-Earth). Also, we have added this sentence to express the point you raise here "The spatial patterns seem to be very RCM dependent, with limited influence of biases in the driving GCM."*

Fig. 5 Aren't the colorbars in Fig.5 switched?

*Yes, thank you for pointing this out*

L464 If Figure 5 is based on bias-corrected data and Fig.6 is not, they cannot be compared.

*Neither figure involves bias adjusted data.*

Fig.6 and S9 RX1day: I noticed different values for MESAN, which seems to be closer to E-OBS in Fig.S9, was there a bias correction performed for the reanalysis products towards E-OBS? I think I miss that explanation, which should be included in the methods. This comment also applies to wind extremes.

*Yes, see answer to comment 1.*

Fig.6 and S9. Why are the Alps not shown for wind (also in Fig.9)?

*The Alps have now been added. (They were originally excluded because in the CORDEX analysis models differed according to whether they simulated fast or slow winds over the Alps relative to everywhere else, suggesting that there were large inconsistencies in how winds were dealt with over mountains. However, since the UPSCALE analysis is based on only one model, wind will be treated in a more consistent way between simulations, and therefore we feel it is ok to add this panel as suggested.)*

L565-569 The analogues analysis is very interesting. One question about analogues recognition: how do you set that the analogue is a good one? I mean, by looking at the correlation of spatial patterns you can always find analogues which can be more or less similar to the target situation, but did you set a correlation limit below which there is not a good analogue for one day? Or did you always find high correlations?

*For each day we choose the best analogue (i.e with the highest correlation coefficient). Most of the time this coefficient was over 0.7 (mostly above 0.8). The day with the least good best analogue has a correlation coefficient of ~0.5. For the larger domain used for temperature correlation coefficients were higher. We did not set a lower limit.*

L574-577 This approach of smoothing before calculating the analogues does not seem to be right. The analogue day should be obtained with the same criteria for all variables/indices, i.e. a given atmospheric circulation is related to a value of temperature, precipitation and wind speed. Calculating it differently brings inconsistent variables. Moreover, this approach seems to be responsible of the overestimation of the return periods of Tx5day, later in lines 601-603, where it is also said that doing it differently results shift downwards.

*The results forTx5day have now been replaced by the versions that do the smoothing last thing (i.e. the analogues are calculated and the u-chronic dataset constructed using daily data, and then the u-chronic dataset is smoothed at the end). This shifts the curves downwards underneath the ones using the original data.*

L603 “but otherwise gives the same results”, isn’t “otherwise” the way it is done in the paper (i.e. first averaging, second analogues)? Then of course it produces the same results as shown. This sentence needs some rephrasing.

*It seems that it was ambiguous what “otherwise” referred to. I meant that apart from being shifted downwards, the results doing the averaging last are the same as the results doing it first (i.e. the relationship between the different curves is the same). I have now rephrased the text (which also takes into account the changes done in response to the previous comment): “For the 5-day variables (Rx5day and TXx5day) the u-chronic dataset was smoothed using a 5-day running mean at the end of the process. We also tried smoothing the daily geopotential height, precipitation and temperature datasets first and then performing the analogue analysis. The relationship between the different curves was largely*

*consistent between the two techniques, but absolute values differed and the shape of the curves changed a little. Results presented here are based on the first technique."*

L606 It would be nice to have a quantitative value of the downscaling and upscaling effect on the indices, such as the relative change with respect to the self-analogue for a given or several return periods.

*I feel that this would become quite complicated to express in the text given the potentially large number of numbers that would need to be written. I think that the graphs show this much more clearly.*

L604 How are model biases treated in Sect. 4.3? In my view, following the above thoughts on bias correction, mean biases should be removed from the analogue series prior to the indices calculation.

*There is no bias correction involved in section 4.3. See also answer to comment 1 above.*

L618-625 The obtained results should be discussed in the context of other studies which show that RCMs yield systematic reduction of temperature biases compared with the driving GCM (Soerland et al. 2018, https://iopscience.iop.org/article/10.1088/1748-9326/aacc77, where some reasons for this are also given).

*We add the following to the discussion section: "Hot biases over mountains reduced with increased resolution, although the cold bias over Scandinavia was worse in CORDEX than in CMIP5. This amplified Scandinavian cold bias in CORDEX is consistent with the findings of Sørland et a l (2018) for mean summer temperature, although we did not find the same reduction of the warm bias in Eastern Europe in CORDEX as they did, possibly due to differences in the models used.*

L692 I would recommend to mention around here or somewhere in the the discussion the potential benefits of current projects such as the EURO-CORDEX flagship pilot studies, about land use change and convection permitting simulations (Jacob et al. 2020, https://doi.org/10.1007/s10113-020-01606-9, for an overview on EURO-CORDEX perspectives).

*A final paragraph has been added: "Finally, ongoing projects such as HighResMIP for CMIP6 (Haarsma et al., 2016), and the CORDEX Flagship Pilot Studies, particularly the FPS on Convective Phenomena at High Resolution over Europe and the Mediterranean (Coppola et al., 2019; Jacob et al 2020), will enable the benefits of high resolution and its effect on European climate projections to be explored more thoroughly. The former will allow a systematic exploration of the effects of increased resolution for multiple GCMs through coordinated experiments simulating the past and future climate. The latter will include a first of its kind large multi-model ensemble at convective permitting resolution for decadal time slices in the present and future for a large domain covering central Europe and part of the Mediterranean."*

*We do not specifically mention the FPS on land use, since land use is not a focus of our paper.*

Spellings and typos

L69 EURO-CORDEX initiative instead of EUROCORDEX project. Use EURO-CORDEX throughout the manuscript (there are some inconsistencies).

*This has been corrected.*

L69 missing bracket after the reference.

*Fixed.*

L74 Maybe better “coarser” than “less”, since it refers to the resolution.

*We have replaced “less” with “coarser”.*

L164 Isn’t a word missing between “adaptation” and “a downscaled”?

*Yes, the word “of” was missing*

Table S1, caption “their corresponding CORDEX simulations to the left”, shouldn’t it be to the right? When describing the crosses, are they really bold? I would say that those in the “common subset” are those with coloured (not bold) crosses.

*Yes, thank you for pointing this out. And although the crosses for the “common subset” are bold as well as coloured, the colour is easier to see, so we have changed the caption accordingly.*

L215-217 Is then the 0.5º common grid the E-OBS grid? Is so, say it explicitly.

*This sentence has been re-phrased as follows: “We use the 0.5° regular longitude-latitude grid of E-OBS since it is in-between the resolution of the CORDEX models and CMIP5, and is computationally feasible.”*

L221 I would say “The sensitivity of the results to the regridding technique...”, also L223 “sensitive to the regridding technique”, L225 “the regridding technique did not make much difference to the results”; but check with a native English speaker.

*Corrected*

L305 Wouldn’t “CORDEX subset” be “common subset”? Here the differences between the left and right panels are being compared.

*This sentence has been rephrased: “Biases for CORDEX using the whole ensemble are very similar to those for the common subset. For CMIP5 the hot biases over the south-east, and over mountain ranges are stronger when using all simulations compared to the subset.”*

L309 Capitalize Figure, also in other parts of the manuscript.

*done*

L318 What do you mean with “are representative of the subregions”?

*This was perhaps ambiguous and has now been rephrased as follows: Results are shown for Northern, Central and Southern Europe, and* ***are representative of results for the smaller PRUDENCE regions that fall within their boundaries****.*

Fig.2 Caption. British Isles are in the top right panel, not in the top left.

*Thank you for pointing out this mistake.*

L350 “E-OBS”. Homogenize notation along the manuscript: it is sometimes Eobs or E-OBS or EOBS.

*This has been changed to “E-OBS” throughout the text.*

L475 heavier.

*Corrected*

L511 dot missing at the end of the sentence.

*Fixed*

L566 Should “lows” be “flows”?

*No, we mean the low pressure at the surface associated with hot surface temperatures due to heating, expansion and rising of surface air. “Heat low” is a standard term, so we choose to keep it.* E.g. https://link.springer.com/referenceworkentry/10.1007/1-4020-3266-8_95

In all figures of return period in which the region of Scandinavia is included, Scandinavia is badly spelled.

*Thank you for pointing this out. This has been corrected.*

L602 “see Methods” should be “see above”, since the procedure is explained above in the same section.

*This has been corrected, thank you.*

L655 dot missing at the end of the sentence.

*Thanks, now corrected.*

L660 “can overestimate”

*corrected*

References

[revised manuscript text omitted]

**Commented [c1]:** Means now replaced by medians and for the "all" ensembles there is now only one member per model.

[Figure]

**Figure 2: Return period plots for (left) TXx5day, (middle column) Rx1day and (right) annual maximum wind, for CMIP5 and CORDEX for Northern Europe (top row (except top rightleft = British Isles)), Central Europe (middle row) and Southern Europe (bottom row). CMIP5 is shown in grey, CORDEX 0.44° in red and CORDEX 0.11° in blue. Thin lines are individual ensemble members, thick lines are multi model medians;, lighter coloursshades for the full ensembles, and darker coloursshades for the subset of models common to CMIP5, and both CORDEX resolutions. Observational datasetss are shown in black, circles for E-OBS temperature and precipitation and MESCAN wind, triangles for MESAN precipitation and DYNAD wind and crosses for ERA5 wind. Confidence intervals based on bootstrapping are shown with dashed lines for the observational datasetss. The time periods considered are 1970-2005 for TXx5day and Rx1day, and 1979-2005 for wind.**

**Commented [c2]:** Annual maximum wind now uses 6 hourly wind instead of sfcWindmax. The “full ensemble” now only includes one member per model

1669

[Figure]

1670 **Figure 3: As for Figure 1 but for the climatological mean of Rx1day. Units mm.**

**Commented [c3]:** Means replaced by medians. “all” ensembles now only have one member per model

[Figure]

**Figure 4: Climatological mean of annual maximum wind for the period 1979-2005 for (a) ERA5, (b) MESCAN (c) DYNAD, and for the multi model medianan of the common subset of models for (d) CMIP5 and (l) CORDEX 0.11° and their biases with respect to the reanalyses datasets (e-g and m-o). (h-k and p-s) are the same but for the full ensembles of CMIP5 and CORDEX. Units meters per second.**

**Commented [c4]:** WindXx now uses 6 hourly data instead of sfcWindmax. Also, “all” ensembles now use only 1 member per model. We replace multi model means by medians. Biases are now shown

[Figure]

**Figure 5: Climatological mean of TXx5day for the ensemble means of three resolutions of HadGEM3-A (UPSCALE) GCM simulations (left) for the period 1985-2011 and their biases with respect to E-OBS (right). (a) E-OBS, (b, c) N96 (130 km), (d, e) N216 (60 km), (f, g) N512 (25 km). Units °C.**

**Commented [c5]:** Colour bars are now switched the correct way round

[Figure]

1680

**Figure 6: Return period plots for (left) TXx5day, middle column Rx1day and (right) annual maximum wind, for the UPSCALE simulations for (top row) the British Isles, (2nd row) Scandinavia, (3rd row) Central Europe, (4th row) Southern Europe, and (last**

**Commented [c6]:** Scandinavia is now spelt correctly, and the Alps have been added for wind.

**row) the Alps. N96 is shown in grey, N216 in red and N512 in blue. Thin lines are individual ensemble members, thick lines represent ensemble means. Observational datasetss are shown in black, circles for E-OBS and MESCAN, triangles for MESAN and DYNAD, and asterisks for ERA5. Confidence intervals based on bootstrapping are shown with dashed lines for the observationsal datasets. The time periods considered are 1985-2011 for TXx5day, 1989-2010 for Rx1day, and 1986-2011 for wind. NB: in contrast to Figure 2 the curves have not been shifted to have the same mean value (see methods), see Figure S10 for the shifted version.**

[Figure]

**Figure 7: Climatological mean of Rx1day for the ensemble means of three resolutions of UPSCALE (left) simulations for the period 1989-2010 and their biases with respect to E-OBS (middle) and the MESAN reanalysis (right). (a) E-OBS, (b) MESAN (c-e) N96, (f-h) N216, (i-k) N512. Units mm.**

[Figure]

**Figure 8: Climatological mean of annual maximum wind for the ensemble means of three resolutions of UPSCALE (left) simulations for the period 1986-2011 and their biases with respect to the observational datasets ERA5 (left column), MESCAN (middle) and MESAN (right). (a) ERA5, (b) MESCAN (c) DYNAD, (d-g) N96, (h-k) N216, (l-o) N512. Units meters per second.**

[Figure]

**Figure 9: Circulation analogue results. Return period plots for (left) TXx5day, (middle) Rx1day and (right) annual maximum wind for (top) the British Isles, (2nd row) Scandinavia, (3rd row) Central Europe, (4th row) Southern Europe and (5th row) the Alps. Grey represents the N96 self-analogues, blue the N512 self-analogues, red is for N96 circulation with N512 variables (e.g. precipitation) and green is for N512 circulation with N96 variables. Thin lines represent individual ensemble members, thick lines represent the mean across individual ensemble members. Blue dashed line represents the original N512 ensemble mean results like those shown in Figure 6 (although sometimes based on a different time period), and the grey dashed lines represent the equivalent for the N96 simulations. Results for TXx5day are based on the period 1985-2011, Rx1day 1986-2011, and wind 1986-2011.**

**Commented [c7]:** TXx5day replaced with version with smoothing last. Alps added.

---

## Author Response (AR3)

Reviewer 2

I am satisfied with the answers the authors gave me and the additions made to the manuscript. Therefore, I recommend the consideration for publication in Geoscientific Model Development.

I only have a few corrections (lines refer to the the tracked-changes version of the manuscript).

L551 "over orography": shouldn't that be "over complex orography"?

*Corrected*

L668 data "were".

*Corrected*

L676 "we repeated" since "we repeat" since past tense is used in the whole paragraph.

*Corrected*

L838 "repeat the analysis using the MESAN reanalysis as reference" better than "repeat results".

*Edited accordingly*

L855, L925 "RCM-dependent"

*Ok*

L877 "therefore those results are not shown" better than "and so are not shown"

*Corrected*

L110 "at the end of the process" is not necessary, in my view, since you already refer to the u-chronic dataset.

*Edited accordingly*

L1104, L1106 "aproaches" better than "techniques".

*Edited accordingly*

L1119 "downscaling effects generally seem to be the most important" better than "it generally seems to be downscaling effects that are the most important".

*Edited accordingly*

L1129 "smoothing" better than "averaging".

*Ok*

L1170 which "other studies"?

*I have added the example of Prein and Gobiet 2017*

L1198 "reanalysis-based".

*Ok*

[revised manuscript text omitted]